

# Eemian Greenland Surface Mass Balance strongly sensitive to SMB model choice

Andreas Plach[1], Kerim H. Nisancioglu[1,2], Sébastien Le clec'h[3,4], Andreas Born[1,5,6], Petra M. Langebroek[7], Chuncheng Guo[7], Michael Imhof[8,5], and Thomas F. Stocker[5,6]

[1]Department of Earth Science, University of Bergen and Bjerknes Centre for Climate Research, Bergen, Norway
[2]Centre for Earth Evolution and Dynamics, University of Oslo, Oslo, Norway
[3]Laboratoire des Sciences du Climat et de l'Environnement, LSCE/IPSL, CEA-CNRS-UVSQ, Université Paris-Saclay, F-91191 Gif-sur-Yvette, France
[4]Earth System Science and Department Geografie, Vrije Universiteit Brussel, Brussels, Belgium
[5]Climate and Environmental Physics, Physics Institute, University of Bern, Bern, Switzerland
[6]Oeschger Centre for Climate Change Research, University of Bern, Bern, Switzerland
[7]Uni Research Climate and Bjerknes Centre for Climate Research, Bergen, Norway
[8]Laboratory of Hydraulics, Hydrology and Glaciology, ETH Zürich, Zürich, Switzerland

*Correspondence to:* Andreas Plach (andreas.plach@uib.no)

**Abstract.** Understanding the behavior of the Greenland ice sheet in a warmer climate, and particularly its surface mass balance (SMB), is important for assessing Greenland's potential contribution to future sea level rise. The Eemian interglacial, the most recent warmer-than-present period in Earth's history approximately 125,000 years ago, provides an analogue for a warm summer climate over Greenland. The Eemian is characterized by a positive Northern Hemisphere summer insolation anomaly, which introduces uncertainties in Eemian SMB when using positive degree day estimates. In this study, we use Eemian global and regional climate simulations in combination with three types of SMB models — a simple positive degree day, an intermediate complexity, and a full surface energy balance model — to evaluate the importance of regional climate and model complexity for estimates of Greenland SMB. We find that all SMB models perform well under the relatively cool pre-industrial and late Eemian. For the relatively warm early Eemian, the differences between SMB models are large which is associated with the representation of insolation in the respective models. For all simulated time slices there is a systematic difference between globally and regionally forced SMB models, due to the different representation of the regional climate over Greenland. We conclude that both the resolution of the simulated climate as well as the method used to estimate the SMB, are important for an accurate simulation of Greenland's SMB. Whether model resolution or SMB method is most important depends on the climate state and in particular the prevailing insolation pattern. We suggest that future Eemian climate model inter-comparison studies are combined with different SMB models to quantify Eemian SMB uncertainty estimates.

## 1 Introduction

The projections of future sea level rise remain uncertain, especially the magnitude and rate of the contributions from the Greenland Ice Sheet (GrIS) and the Antarctic Ice Sheet (Church et al., 2013; Mengel et al., 2016). In addition to improving dynamical



climate models, it is important to test their ability to simulate documented warm climates. Past interglacials are relevant examples as these were periods of the recent past with relatively stable warm climates persisting over several millennia, providing a benchmark for testing key dynamical processes and feedbacks given a different background climate state. Quaternary interglacials exhibit a geological configuration similar to today (e.g., gateways and topography), and have been frequently used as

analogues for future climates (e.g., Yin and Berger, 2015). In particular, the most recent interglacial, the Eemian (approx. 130 to 116 ka) has been used to better understand ice sheet behavior during a warm climate.

Compared to the pre-industrial, the Eemian is estimated to have had less Arctic summer sea ice, warmer Arctic summer temperatures, and up to 2°C warmer annual global average temperatures (CAPE Last Interglacial Project Members, 2006; Otto-Bliesner et al., 2013; Capron et al., 2014). Ice core records from NEEM (the North Greenland Eemian Ice Drilling project

in northwest Greenland) indicate a local warming of 8.5±2.5°C (Landais et al., 2016) compared to pre-industrial levels. While, total gas content measurements from the deep Greenland ice cores GISP2, GRIP, NGRIP, and NEEM indicate that the Eemian surface elevation at these locations was no more than a few hundred meters lower than present (Raynaud et al., 1997; NEEM community members, 2013). Proxy data derived from coral reefs show a global mean sea level at least 4 m above the present level (Overpeck et al., 2006; Kopp et al., 2013; Dutton et al., 2015).

Several studies have investigated the Eemian GrIS. Nevertheless, there is no consensus on the extent to which the GrIS retreated during the Eemian. Scientists have applied ice core reconstructions (e.g., Letréguilly et al., 1991; Greve, 2005), Global Circulation Models (GCMs) of various complexities (e.g., Otto-Bliesner et al., 2006; Stone et al., 2013) combined with regional models (e.g., Robinson et al., 2011; Helsen et al., 2013) to create Eemian temperature and precipitation forcing over Greenland. Based on these reconstructed or simulated climates, different models have been used to calculate the Surface

Mass Balance (SMB) over Greenland for the Eemian. The vast majority of these use the Positive Degree Day (PDD) method introduced by Reeh (1989), which is based on an empirical relation between melt and temperature. PDD has been shown to work well under present-day conditions (e.g., Braithwaite, 1995) and has been widely used by the community due to its simplicity and ease of integration with climate and ice sheet models. More recent studies employ physically-based approaches to calculate the SMB, ranging from empirical models (e.g., Robinson et al., 2010) to Surface Energy Balance (SEB) models

(e.g., Helsen et al., 2013).

It is important to note that the relatively warm summer (and cold winter) in the Northern Hemisphere during the early Eemian (130-125 ka) was caused by a different insolation regime compared to today, not due to increased concentrations of greenhouse gases (GHGs) which is the reason for the recent observed global warming (e.g., Langebroek and Nisancioglu, 2014). The early Eemian was characterized by a positive solar insolation anomaly during Northern summer caused by a higher

obliquity and eccentricity compared to present, as well as a favourable precession giving warm Northern Hemisphere summers at high latitudes (Yin and Berger, 2010). The higher summer insolation over Greenland, compared to today, adds snow/ice melt which is not included in the PDD approach (e.g., Van de Berg et al., 2011). These limitations should be kept in mind when using past warm periods as analogues for future warming (e.g., Ganopolski and Robinson, 2011; Lunt et al., 2013). However, the higher availability of proxy data compared to preceding interglacials makes the Eemian a better candidate to

investigate warmer conditions over Greenland (Yin and Berger, 2015). Furthermore, the amplification of summer warming



over Greenland has been found to be effective regardless of whether the warming is caused by higher insolation or increased GHGs concentrations (Masson-Delmotte et al., 2011).

In this study we assess the importance of the representation of small-scale climate features and the impact of SMB model complexity (i.e., using three SMB models) when calculating the SMB for warm climates such as the Eemian. High-resolution pre-industrial and Eemian Greenland climate is provided by downscaling global time slice simulations with the Norwegian Earth System Model (NorESM) using the Regional Climate Model (RCM) MAR (Modèle Atmosphérique Régional). Based on these global and regional climate simulations three different SMB models are applied, including (1) a simple, empirical PDD model, (2) an intermediate complexity SMB model explicitly accounting for solar insolation, as well as (3) the full Surface Energy Balance (SEB) model implemented in MAR.

The models, data, and experimental design are described in Sec. 2, followed by a review of previous Eemian GrIS studies in Sec. 3. The results of the pre-industrial and Eemian simulations are presented in Sec. 4 and 5, respectively. The challenges and uncertainties are discussed in Sec. 6. Finally, a summary of the study is given in Sec. 7.

## 2    Models and methods

### 2.1    Model description

We use the output of an Earth System Model (ESM) and a Regional Climate Model (RCM) to assess the influence of model resolution on the simulated SMB over Greenland. The regional model is forced with the output from the global model at its boundaries (i.e., the regional model is constrained by the global model simulations). Furthermore, we test three different SMB models of various complexity, all forced with the simulated global and the regional climates. Throughout this study we refer to the two simulated climates as global (from the ESM) and regional (from the RCM).

**Norwegian Earth System Model (NorESM)**

The Norwegian Earth System Model (NorESM) was first introduced by Bentsen et al. (2013) and included as version NorESM1-M in phase 5 of the Climate Model Intercomparison Project (CMIP5; Taylor et al., 2011). NorESM is based on the Community Climate System Model version 4 (CCSM4; Gent et al., 2011) but was modified to include an isopycnic coordinate ocean general circulation model that originates from the Miami Isopycnic Coordinate Ocean Model (MICOM; Bleck et al., 1992), an atmosphere component with advanced chemistry-aerosol-cloud-radiation schemes known as the Oslo version of the Community Atmosphere Model (CAM4-Oslo; Kirkevåg et al., 2013), and the HAMburg Ocean Carbon Cycle (HAMOCC) model (Maier-Reimer, 1993; Maier-Reimer et al., 2005) adapted to the isopycnic ocean model framework.

In this work, we use a newly established variation of NorESM1-M, named NorESM1-MF (Guo et al., in prep.), that retains the resolution (2-degree atmosphere/land, 1-degree ocean/sea ice) and overall quality of NorESM1-M, but is a computationally efficient configuration that is designed for multi-millenial and ensemble simulations. In NorESM1-MF, the model complexity is reduced by replacing CAM4-Oslo with the standard, prescribed aerosol chemistry of CAM4. The coupling frequency between



atmosphere–sea ice and atmosphere–land is reduced from half-hourly to hourly, and the dynamic sub-cycling of the sea ice is reduced from 120 to 80 sub-cycles. These changes speed up the model by ~30%, while having a relatively small effect on the model's overall climate. In addition, some recent code developments for NorESM CMIP6 are implemented, as documented in detail by Guo et al. (in prep.). Especially, the updated ocean physics in NorESM1-MF leads to improvements over NorESM1-

M in the simulated strength of the Atlantic Meridional Overturning Circulation and Arctic sea ice, both of which are important metrics when simulating past and future climates.

**Positive Degree Day (PDD) model**

The Positive Degree Day (PDD) method was introduced by Reeh (1989). The model is based on an empirical relationship between near surface temperature and surface melt. Its minimum requirements are monthly near surface temperature and total

precipitation. Due to its simplicity and low input requirements, it is often used in paleo studies where data availability is limited and the timescales of interest are long. Here, we use the PDD model as a legacy baseline with commonly used melt factors for snow and ice (e.g., Letréguilly et al., 1991; Ritz et al., 1997; Lhomme et al., 2005; Born and Nisancioglu, 2012).

   The model integrates the number of days with temperatures above freezing into a PDD variable, which is multiplied by empirically based melt factors to calculate the amount of snow and ice melt. Different factors for snow and ice are applied to

account for differences in albedo. The temperature variability during a month is simulated assuming a Gaussian distribution. The most important PDD model parameters are summarized in Tab. 1. Since a PDD model exclusively uses temperature to calculate melt, it only accounts for terms in the surface energy balance which are directly related to temperature. It does not directly account for shortwave radiation, which means that a PDD model is always tuned to present-day insolation conditions. This is of particular relevance in studies of past climates, such as the Eemian, which exhibit different seasonal insolation

patterns compared to today. Van de Berg et al. (2011) show that a PDD model underestimates melt compared to a full SEB model, when using PDD melt factors tuned to present-day conditions.

   Here, we use a PDD model introduced by Seguinot (2013) and modify it to our needs. The PDD model uses the total monthly precipitation and calculates the snow fraction and accumulation via two threshold temperatures. If the temperature is below -10°C all precipitation falls as snow, and if the temperature is above 7°C all precipitation falls as rain and does not contribute

to the accumulation. In between these extremes, a linear relation is applied to calculate the snow fraction.

**Ber/ge/n Snow Simulator (BESSI)**

The intermediate complexity SMB model, Ber/ge/n Snow Simulator (BESSI) is designed to be computationally efficient and to be forced by low complexity climate models. It uses only daily mean values of three input fields, temperature, precipitation and downward shortwave radiation. Furthermore, outgoing longwave radiation is calculated prognostically while incoming

longwave radiation is calculated with a Stefan-Boltzmann law using the input surface air temperature and a globally constant air emissivity. BESSI is introduced in Imhof (2016); Born et al. (in prep.). It is a physically consistent multi-layer SMB model with firn compaction. The firn column is modeled on a mass-following, Lagrangian grid. BESSI uses a surface energy balance that includes heat diffusion in the firn, retention of liquid water, and refreezing. The only process it neglects is sublimation



**Table 1.** Model parameters of the empirical PDD model

| PDD model parameters | |
| --- | --- |
| PDD snow melt factor | 3 mm/PDD |
| PDD ice melt factor | 8 mm/PDD |
| maximum snow refreeze | 60 % |
| maximum ice refreeze | 0 % |
| all snow temperature | -10°C |
| all rain temperature | 7°C |
| standard deviation of the near surface temperature | 4.5 °C |

**Table 2.** Model parameters of the intermediate complexity model BESSI

| BESSI model parameters | |
| --- | --- |
| albedo dry snow | 0.85 |
| albedo wet snow | 0.70 |
| albedo ice | 0.40 |
| bulk coefficient sensible heat flux | 5.0 W/m$^2$/K |
| air emissivity | 0.87 |
| pore volume available to liquid water | 0.1 |
| number of snow layers | 15 |

which is of low importance for the mass balance of Greenland. Firn densification is realized with models commonly used in ice core research, following Herron and Langway (1980) for densities below 550 kg/m$^3$ and Barnola et al. (1990) for densities above 550 kg/m$^3$. There is no water routing on the surface, but the firn can hold up to 10% of its pore volume in water. All excess water percolates into the next grid box below and if it reaches the bottom of the firn layer it is removed from the system.

5 Tab. 2 summarizes the most important BESSI model parameters.

**Modèle Atmosphérique Régional (MAR)**

We use the Modèle Atmosphérique Régional (MAR) to produce high resolution SMB over the GrIS during the Eemian period. MAR is a regional atmospheric model fully coupled to the land surface model SISVAT which includes a detailed snow energy balance model (Gallée and Duynkerke, 1997). The atmospheric part of MAR uses the solar radiation scheme of Morcrette

10 et al. (2008) and accounts for the atmospheric hydrological cycle (including a cloud microphysical model) based on Lin et al. (1983) and Kessler (1969). The snow-ice part of MAR is derived from the snowpack model Crocus (Brun et al., 1992). This





1-D model simulates fluxes of mass and energy between snow layers, and reproduces snow grain properties and their effect on surface albedo.

The present work uses MAR version 3.6 in a similar model setup as in Le clec'h et al. (2017) with a fixed present-day ice sheet topography. We use a horizontal resolution of 25 x 25 km covering the Greenland domain (6600 grid points; Stereographic Oblique Projection with its origin at 40°W and 70.5°N) from 60° W to 20° W and from 58° N to 81° N. The model has 24 atmospheric layers from the surface to an altitude of 16 km. SISVAT has 30 layers to represent the snowpack (with a depth of at least 20 m over the permanent ice area) and 7 levels for the soil in the tundra area. The snowpack initialization is described in Fettweis et al. (2005).

MAR has often been validated against in situ observations, e.g., in Fettweis (2007); Fettweis et al. (2013, 2017). Lateral boundary conditions can be provided either by reanalysis datasets (such as ERA-interim or NCEP) to reconstruct the recent GrIS climate (1900-2015) (Fettweis et al., 2017) or by GCMs (e.g., Fettweis et al., 2013). In this study, the initial topography of the GrIS as well as the surface types (ocean, tundra and permanent ice) are derived from Bamber et al. (2013). At its lateral boundaries, MAR is forced every 6 hours with atmospheric fields (temperature, humidity, wind and surface pressure) and at the ocean surface, sea surface temperature and sea ice extent from the NorESM output are prescribed. For this all NorESM output is linearly interpolated on the 25 x 25 km MAR grid.

For the SMB calculation, MAR assumes ice coverage after all firn has melted. The calculated SMB is weighted by a ratio-of-glaciation mask derived from Bamber et al. (2013). For consistency, this mask is used for all PDD- and BESSI-derived SMBs as well. Regions with less than 50% permanent ice cover are not considered for our analysis (same as Fettweis et al., 2017).

## 2.2 Experimental design, model spin-up and terminology

**Model experiment setup**

We use five NorESM time slice simulations, a pre-industrial control run and four runs representing Eemian conditions at 130, 125, 120, and 115 ka. All five NorESM runs are dynamically downscaled with MAR, i.e. MAR is constrained with NorESM output at its boundaries. All five runs from NorESM and MAR are used to force different SMB models.

The NorESM pre-industrial experiment is spun up for 1000 years to reach a quasi-equilibrium state, followed by another model run of 1000 years representing the pre-industrial control simulation. The four Eemian time slice experiments (130, 125, 120, 115 ka) are branched off after the 1000 years spin-up experiment and run for another 1000 years each. The simulations are close to equilibrium at the end of the integration, with very small trends in, e.g., top of the atmosphere radiation imbalance (0.02, 0.04, 0.02, 0.02 W/m$^2$ per century, respectively, between model years 1801-2000; all trends are statistically not significant) and global mean ocean temperature (-0.008, -0.01 -0.03, -0.03 K per century, respectively, between model years 1801-2000; all trends are statistically significant except for the 130 ka case). Statistical significance of the calculated trends is tested using the Student's $t$-test with the number of degrees of freedom, accounting for autocorrelation, calculated following Bretherton et al. (1999). Trends with $p$ values < 0.05 are considered to be statistically significant.



**Table 3.** Greenhouse concentrations and orbital parameters used for the NorESM and MAR climate simulations (PI...pre-industrial)

|  | 130 ka | 125 ka | 120 ka | 115 ka | PI |
|---|---|---|---|---|---|
| $CO_2$ [ppm] | 257.0 | 276.0 | 269.0 | 273.0 | 284.7 |
| $CH_4$ [ppb] | 512.0 | 640.0 | 573.0 | 472.0 | 791.6 |
| $N_2O$ [ppb] | 239.0 | 263.0 | 262.0 | 251.0 | 275.7 |
| CFC-11 [ppt] | 0 | 0 | 0 | 0 | 12.5 |
| CFC-12 [ppt] | 0 | 0 | 0 | 0 | 0 |
| Eccentricity | 0.0382 | 0.0400 | 0.0410 | 0.0414 | 0.0167 |
| Obliquity [deg] | 24.24 | 23.79 | 23.12 | 22.40 | 23.44 |
| Long. of perih. [deg] | 228.32 | 307.13 | 27.97 | 110.87 | 102.72 |

The model configurations follow the protocols of the third phase of the Paleoclimate Modeling Intercomparison Project (PMIP3). Compared with the experimental setup of the pre-industrial control simulation, only orbital forcing and greenhouse gas concentrations are changed in the four Eemian experiments. The greenhouse gas concentrations and orbital parameters used for the five time slice experiments (NorESM as well as MAR) are listed in Tab. 3.

For the MAR experiments, NorESM is run for another 30 years for each of the five experiments and the output is saved 6-hourly. This 30 years are used as boundary forcing for MAR. The first four years are disregarded as spin-up and the final 26 years are used for the analysis here.

    BESSI is forced with daily fields of temperature, precipitation, and downwards shortwave radiation of these final 25 climate model years of NorESM and MAR respectively. The forcing is applied cyclically (forwards and backwards) 6 times until SMB
values reach an equilibrium. The SMBs of the final $7^{th}$ cycle are used to calculate annual means over 25 years which are used in the analysis.

**Experiment terminology**

We force the PDD model with monthly temperature and precipitation fields from NorESM and MAR respectively, and refer to the resulting SMBs as NorESM-PDD and MAR-PDD respectively. MAR has a full surface energy balance (SEB) model
implemented and its derived SMB is refered to as MAR-SEB. Additionally, we force the intermediate complexity SMB model, BESSI, with daily NorESM and MAR temperature, precipitation and the downward shortwave radiation, and call its output NorESM-BESSI and MAR-BESSI, respectively. An overview of the experimental design is shown in Fig. 1.

    For lack of observational data with a comprehensive coverage, we use the most complex model, MAR-SEB, as our reference SMB model. The standard PDD experimental setup (see Tab. 1) is tuned to present-day Greenland. The intermediate complexity
SMB model, BESSI, is tuned to the MAR-SEB under pre-industrial conditions in terms of SMB and refreezing. The first tuning goal is the total integrated SMB within +/- 50 Gt and the smallest possible Root Mean Square (RMS) error. From the set of





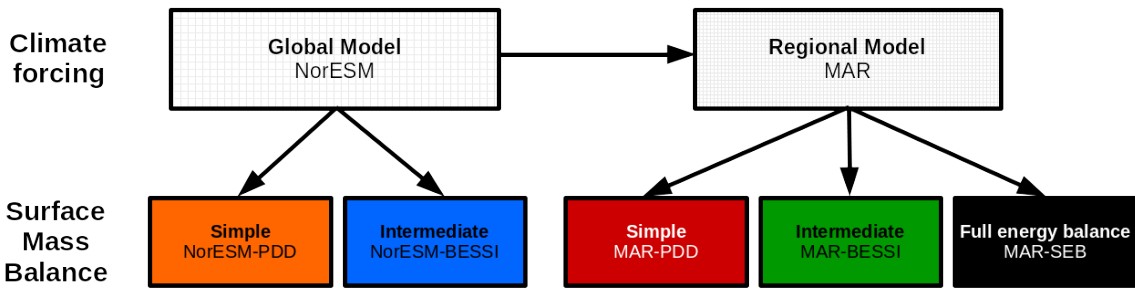

**Figure 1.** Overview of the experimental design. The simple PDD and intermediate BESSI SMB simulations are forced with output of our global climate (from NorESM) and the regional climate (from MAR). Additionally a SMB is derived from the SEB model implemented in MAR. This flow of experiments is performed in the same way for all five time slices (130, 125, 120, 115 ka, pre-industrial).

model parameters which fullfill this goal we choose the set which showed the best fit of refreeze (total amount and RMS error). The most important model parameters for the empirical PDD and the intermediate SMB model, BESSI, are summarized in Tab. 1 and 2, respectively.

We compare the five SMB model experiments (NorESM-PDD, MAR-PDD, NorESM-BESSI, MAR-BESSI, and MAR-SEB)
under pre-industrial conditions and analyze the evolution of their respective SMB components during the Eemian interglacial.

**Interpolation of temperature fields to a higher resolution grid**

To derive realistic near surface temperatures on a higher-than-climate-model resolution (e.g., an ice sheet model grid, but also from NorESM to MAR grid) it is necessary to account for the coarse topography in the initial climate model. In this study, the NorESM temperature is bilinearly interpolated on the MAR grid and a temperature lapse rate correction is applied to account
for the height difference caused by different resolutions.

The model topographies of Greenland in NorESM and MAR are shown in Fig. 2 (panel a and c). Both represent the present-day ice sheet, but in different spatial resolutions. The difference to the observed, high-resolution topography (Schaffer et al., 2016) is also shown in Fig. 2 (panel b and d). Due to the lower spatial resolution of NorESM and the resulting smoothed model topography differences between model and observations are large and cover extensive areas. On the contrary, the differences
for the MAR topography to observations are localized at the margins of the ice sheet and much smaller. The strong resemblance of the MAR topography and observations allows us to use MAR temperature directly, without any correction. Furthermore, we perform a sensitivity test for PDD-derived SMB comparing various temperature lapse rates and discuss its results in Sec. 4.2.



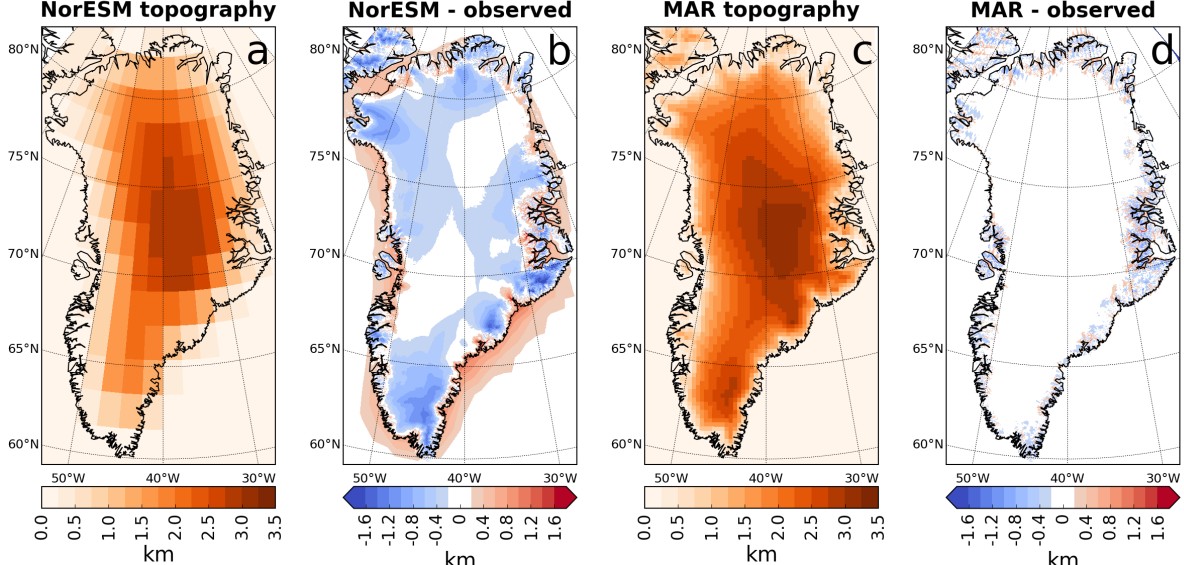

**Figure 2.** Greenland model topographies and differences to observed Greenland topography from Schaffer et al. (2016). a) NorESM Greenland topography (on original resolution of 1.9 x 2.5°(latitude/longitude)) b) NorESM minus observed c) MAR Greenland topography (on original resolution of 25 x 25 km) d) MAR minus observed.

## 3 Comparison of previous Eemian Greenland studies

Scientists started modeling the Eemian GrIS more than 25 years ago (Letréguilly et al., 1991). However, a clear picture of the minimum extent and shape of the GrIS during this critical period of the past is still missing. The estimated contributions of the GrIS to Eemian sea level rise differ largely and vary between 0.4 and 5.6 m. An overview of previous studies and their

estimated Eemian sea level rise from Greenland is given in Fig. 3.

Early studies use Eemian temperature anomalies derived from ice core records and perturb a present-day temperature field in order to get estimated Eemian temperatures over Greenland. This index method is based on single Greenland ice cores (Letréguilly et al., 1991; Ritz et al., 1997) or a composite of ice cores from Greenland and Antarctica (Cuffey and Marshall, 2000; Huybrechts, 2002; Tarasov and Peltier, 2003; Lhomme et al., 2005; Greve, 2005). All these "index studies" employ a

present-day precipitation field when modeling Greenland during the Eemian. In the mid-2000s scientists started using climate models to simulate Eemian climate. The first studies use GCM output directly to force SMB models (Otto-Bliesner et al., 2006; Fyke et al., 2011; Born and Nisancioglu, 2012; Stone et al., 2013). Later studies use statistical (Robinson et al., 2011; Calov et al., 2015) and dynamical downscaling of GCM simulations (Helsen et al., 2013) to create climate input for SMB models. Quiquet et al. (2013) use an adapted index method employing a Eemian temperature and precipitation anomalies from two

GCMs.



To estimate the Eemian ice sheet extent and volume changes these studies use various ice sheet models. However, all used ice sheet models that are based on similar ice flow equations, either the Shallow Ice Approximation (SIA) or a combination of SIA and the Shallow Shelf Approximation (SSA), i.e., the choice of ice sheet models can not explain the differences between the studies. Therefore, we are not discussing the ice dynamics used further. For more details on ice dynamic approximations

see Greve and Blatter (2009). Here, we focus on the choice of climate forcing and calculation of SMB.

The studies using climate models apply different strategies to account for climate-ice sheet interaction. The early studies employ a one-way coupling by forcing the ice sheet model with an Eemian climate without any feedback between the ice sheet and the climate (Otto-Bliesner et al., 2006; Fyke et al., 2011; Born and Nisancioglu, 2012; Quiquet et al., 2013). Later studies use more advanced coupling by performing GCM simulations with various Eemian ice sheet topographies and interpolating in

between the different GCM states according to the evolution of the ice sheet model (Stone et al., 2013) or changing the GrIS topography in RCM simulations every 1.5 ka following the topography evolution in an ice sheet model (Helsen et al., 2013).

The SMB in most of the previous studies of the Eemian is calculated with the empirical PDD model. The exceptions are Robinson et al. (2011) and Calov et al. (2015) who use an intermediate complexity statistical downscaling with an adapted PDD scheme to also include shortwave radiation. Furthermore, Helsen et al. (2013) use a full surface energy balance model

(included in a RCM). While Goelzer et al. (2016) employ a fully-coupled (coarse resolution) GCM-ice sheet model to simulate the evolution of the GrIS during the Eemian, they also use a PDD model to calculate the SMB.

A comparison of the minimum Greenland ice sheet shapes and extents during the Eemian, as simulated by several studies, is shown in Fig. 4. The estimated ice sheet extent and the volume loss (expressed as sea level rise contribution) vary strongly between studies. All models show large ice loss in the southwest of the ice sheet, and several studies show a separation of the

ice sheet into a northern and a southern dome. Additionally, some studies also exhibit extensive ice loss in the north, while others almost show no retreat there. Overall, the estimated Eemian sea level rise contribution from Greenland remains uncertain due to the big differences between these studies. It is important to emphasize that the early studies did not have the same proxy data (i.e. ice core records) available to constrain their models, as in the more recent studies. As an example, Otto-Bliesner et al. (2006) assume an ice-free Dye-3 location during the Eemian as an evaluation criteria for their simulations. However, scientists

now argue that there is indeed Eemian ice at the bottom of all deep ice core sites (Johnsen and Vinther, 2007; Willerslev et al., 2007).

## 4   Pre-industrial simulation results

### 4.1   Pre-industrial climate

The pre-industrial annual mean NorESM and MAR temperatures are compared with the observations in Fig. 5 (top row). The

observations are taken from a collection of shallow ice core records and coastal weather station data compiled by Faber (2016). The data covers the time period from 1890 to 2014. However, individual stations cover only parts of this period. DMI_1 stations provide annual mean temperature and precipitation whereas DMI_2 stations only provide temperature. The NorESM temperature is bilinearly interpolated to the MAR grid and corrected to the MAR topography with a model consistent, temporally and



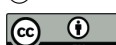

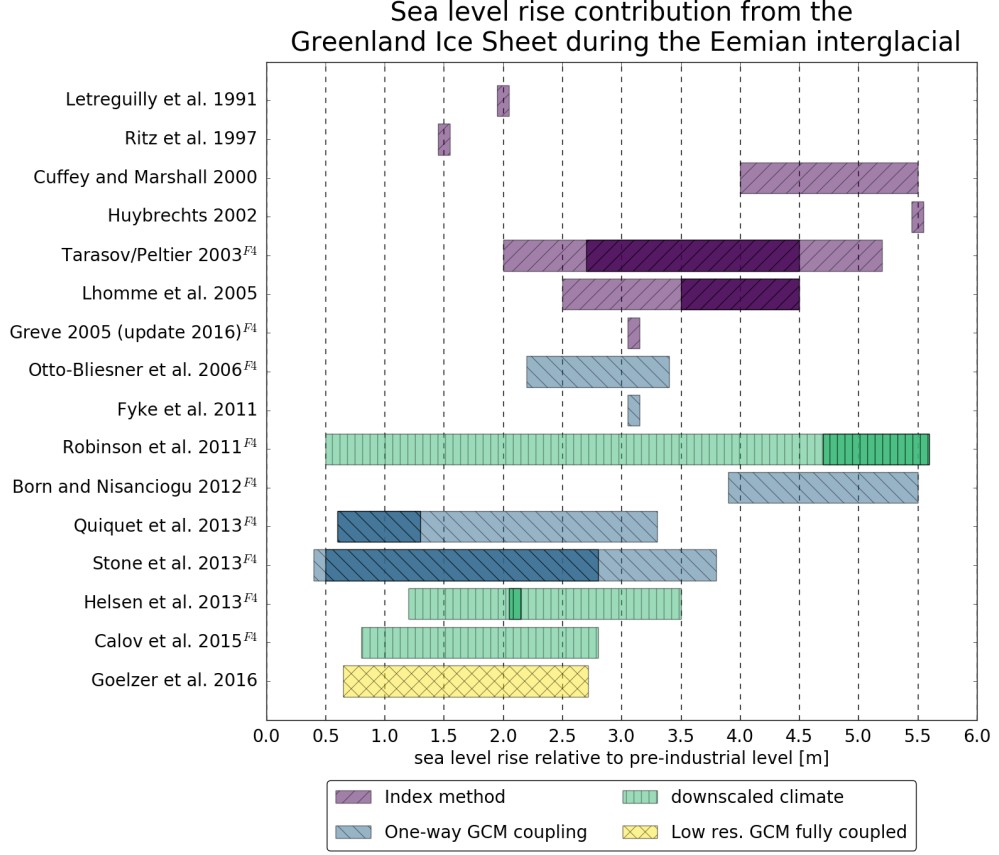

**Figure 3.** Overview of previously published GrIS contributions to the Eemian sea level high stand. The studies are color-coded according to the atmospheric forcing used. More likely values are indicated with darker colors if provided in the respective studies. Different conversions from melted ice volume to sea level rise are used and therefore the contributions are transformed to a common conversion if sufficient data (i.e. pre-industrial ice volume for the respective simulations) is available. Due to this conversion some of the values in this figure are slightly different from the original publications. We use a simple uniform distribution of the water volume on a spherical Earth. The common sea level rise conversion is performed for Greve (2005), Robinson et al. (2011), Born and Nisancioglu (2012), Quiquet et al. (2013), Helsen et al. (2013), and Calov et al. (2015). The minimum ice extent and topography of studies marked with *F4* are shown in Fig. 4

spatially varying lapse rate derived from NorESM. Sensitivity experiments with various lapse rates are discussed in Sec. 4.2. Due to a lack of observations, we are not comparing the exact same period here, resulting in an inherent offset between climate model and observations.

NorESM and MAR temperatures agree well with the observations from the coastal regions. MAR simulates warmer tem-
5   peratures than NorESM at the northern rim of Greenland, an area which is underrepresented in the observations. The cold temperatures in the interior are better captured by MAR than by NorESM. The total NorESM and MAR precipitation, under pre-industrial conditions, is shown in Fig. 5 (bottom row). Compared to the observations, both climate models overestimate





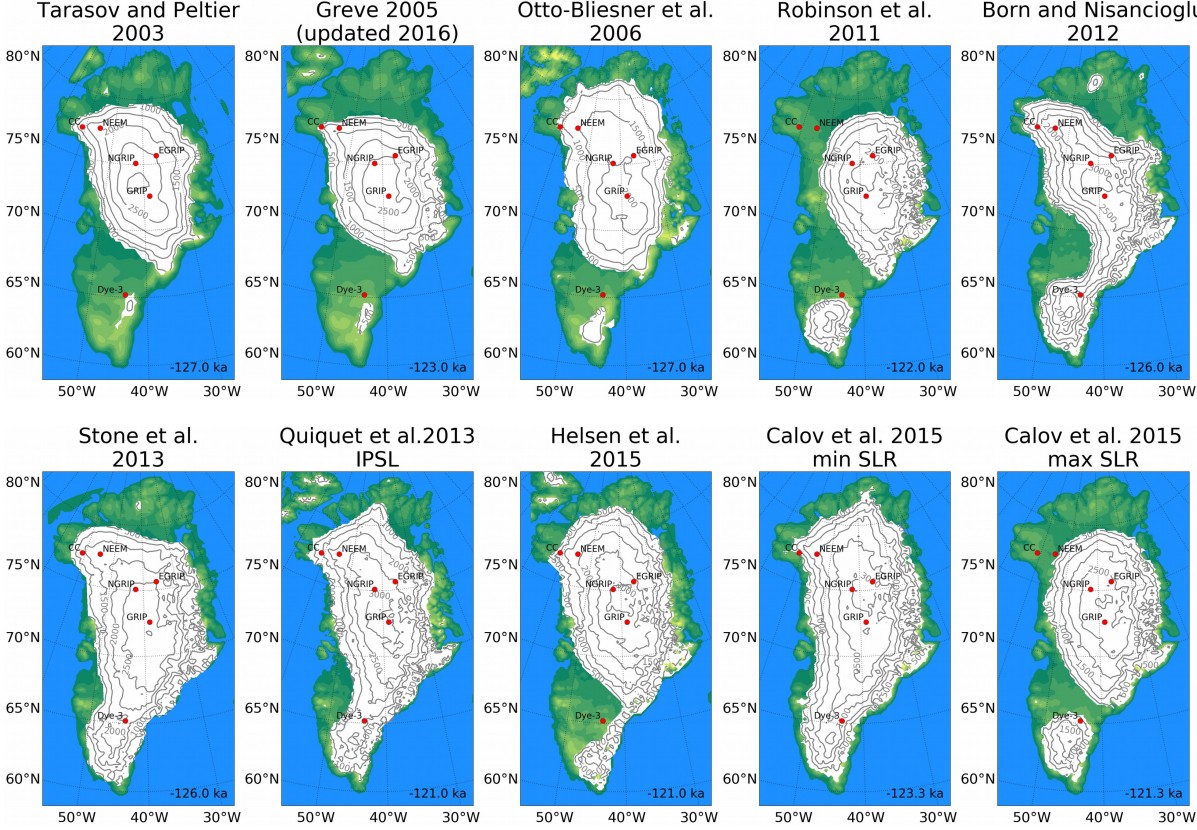

**Figure 4.** Overview of previously modeled minimum ice extent and topography of the Eemian GrIS. The number in the lower right corner of each panel refers to the timing of the minimum ice extent in the respective simulation. Deep ice core locations are indicated with red circles.

precipitation. This overestimation is visible due to the fact that most scatter points are above the gray 1:1 diagonal, indicating a too high model value. However, it is important to note that observations from ice cores represent accumulation (i.e. precipitation minus snow drift, sublimation, and similar processes) rather than precipitation, which can partly explain an overestimation at the ice core locations. The MAR precipitation shows less spread and is closer to the observations than NorESM. The pre-
5  cipitation pattern of NorESM is related to its coarse representation of the topography. MAR on the other hand resolves coastal and local maxima. Unfortunately, the locations with the highest precipitation rates are not covered by the observations.

## 4.2 Sensitivity of PDD-derived SMB to temperature lapse rate correction

To compare the temperature fields, which are computed at different model grid resolutions, a temperature lapse rate correction is applied accounting for the elevation difference of the model surfaces. Often spatially uniform values between 5 (e.g., Abe-
10  Ouchi et al., 2007; Fyke et al., 2011) and close to 8°C/km (e.g., Huybrechts, 2002) are used. Temporally varying temperature lapse rates are used by Quiquet et al. (2013) and Stone et al. (2013). We use 6.5°C/km as our default lapse rate (e.g., Born



and Nisancioglu, 2012). However, we test spatially and temporally uniform lapse rates between 5 and 10°C/km. Additionally, we derive the lapse rate of the free troposphere from the NorESM vertical atmospheric air column above each grid cell (i.e., minimum lapse rate above the surface inversion layer). We refer to this as the *3D lapse rate*. Furthermore, we calculate the *moist adiabatic lapse rate* (MALR; American Meteorological Society, 2018) from the thermodynamic state of the NorESM

surface air layer via pressure, humidity and temperature. The MALR is the rate of temperature decrease with height along a moist adiabat. Both, the 3D lapse rate and the MALR, vary in time and space.

The integrated PDD-derived SMB over Greenland, using these different lapse rates, is compared in Fig. 6. Greenland is split into four sectors along 72.5° N and 42.5° W to investigate regional differences. We first focus on the temporally and spatially uniform lapse rates, shown in red colors. Overall, the different lapse rates have little effect on the PDD-derived SMB. The

extremely high lapse rate of 10°C/km shows the strongest reduction in SMB. The regional contributions do not change much, except in the SE sector: higher lapse rates give lower SMB in southeast Greenland. For the uncorrected temperature fields of NorESM (gray columns), the relative contribution of the SE and SW sectors of Greenland are switched, giving a larger SMB contribution from SE Greenland. In this uncorrected case, ablation is almost completely absent in the SE sector, even in the lowest coastal regions (not shown), which is not realistic (compare our reference MAR-SEB results in Fig. 7e). Furthermore,

the ablation in the SW reaches much lower values than our reference MAR-SEB results.

The general pattern for the PDD-derived SMB fields, calculated using a uniform temperature lapse rate, is that the SMB is reduced as the lapse rate increases, mainly due to the decrease in SMB in SE Greenland. This might seem counter intuitive, since most of the NorESM topography is lower than observations (blue colors in Fig. 2). However, a closer look at Fig. 2 reveals that large parts of the margins are higher than observations (red colors) which results in a warming when applying the

lapse rate correction. Additionally, the margins are also the major melt regions. Therefore, higher lapse rates lead to warmer margins, and as a result, to lower SMB.

Both, the 3D lapse rate and the MALR corrections (blue colors) lead to SMB values (total and regional contributions) which lie between what follows from using 6.5 and 8°C/km as uniform lapse rates. This makes sense since the mean values of the 3D lapse rate and MALR are close to 6.5 and 8°C/km, respectively. Not just the total SMB, but also the spatial pattern of the SMB

over Greenland is similar with all the lapse rate corrections (not shown). Only the SMB derived with uncorrected temperatures shows a different pattern – the contributions from the SE and SW are switched, i.e. there is more extensive melt in the SW and less in the SE because the coastal small-scale features are absent in the uncorrected NorESM temperature due to its relatively coarse resolution.

We conclude that it is necessary to apply a temperature lapse rate correction to lower resolution temperature fields to obtain

a realistic spatial SMB pattern. Using GCM temperature directly in a PDD model results in a coarse representation of the SMB – a wide ablation zone in the west and virtually no ablation on the east coast (not shown). However, the exact value of the lapse rate is less important when using a PDD model. For the comparison of NorESM temperature and observations in Fig. 5 we use the model consistent 3D lapse rate.

The influence of the lapse rate correction on the PDD-derived SMB is minimal and the results from the 3D lapse rate and

the uniform 6.5°C/km (which was used before) are very similar, therefore are we using the latter in our PDD calculations. We



do not aim to adapt PDD in this study but rather use it as a legacy baseline. The correction is applied to NorESM-PDD and NorESM-BESSI. MAR temperature is not corrected, since the MAR topography represents observations sufficiently well (see Fig. 2).

### 4.3 Pre-industrial Surface Mass Balance

The simulated pre-industrial SMB from all five model combinations is shown in Fig. 7. Panel 7e shows our reference, MAR-SEB, which we compare all model experiments to. Both NorESM-derived SMBs, NorESM-PDD and NorESM-BESSI (panel 7a and 7c), show a stronger and spatially more extensive positive SMB anomaly compared to the other experiments. The accumulation in the south looks similar to the NorESM precipitation in Fig. 5, which leads to a positive SMB anomaly. Since NorESM can not resolve the narrow precipitation band in the southeast correctly, the accumulation is spread out over a larger region reaching further inland. The narrow ablation zone in the southeast simulated by MAR-SEB, is much less pronounced in all four simpler model experiments. MAR-PDD and MAR-BESSI also show a positive SMB anomaly on the margins, but not in the southern interior.

Figure 7f shows the Greenland-integrated SMB components. All models are compared on a common ice mask (i.e. less than 50% permanent ice cover in MAR; see Sec. 2.1). The NorESM-forced model experiments, NorESM-PDD and NorESM-BESSI, show the higher total integrated SMBs (gray bars) as a result of the high accumulation (green bars) and low melt (red bars). Both are related to the lower resolution of NorESM (i.e. the narrow precipitation band in the southeast is not captured and the precipitation is smeared over the whole southern tip of Greenland). Furthermore, the lower resolution of NorESM causes its ice mask to reach beyond the common MAR ice mask (not shown); potential NorESM ablation regions are partly cut off. The MAR forced models, MAR-PDD, MAR-BESSI, and MAR-SEB, show better agreement with each other, but the simpler models underestimate melt and refreeze. This is generally true for all four simpler models. In particular, the refreeze values are much lower than in our reference, MAR-SEB. It is not surprising that the PDD model does not capture refreeze as it uses a very simple parameterization (i.e., refreeze is limited to 60 % of the monthly accumulation; following Reeh, 1989). The intermediate model, BESSI, has a firn model implemented (see Sec. 2.1), but also shows much less refreeze than our reference, MAR-SEB.

## 5 Eemian simulation results

### 5.1 NorESM Eemian simulations

Simulated changes of annual mean, boreal winter (December-January-February, DJF), and boreal summer (June-July-August, JJA) near surface temperatures for the four Eemian time slices are shown in Fig. 8. Annual mean temperature changes are relatively small compared to the seasonal changes due to the strong seasonal insolation anomalies during the Eemian interglacial. However, there was a total annual irradiation surplus at high latitudes during the Eemian (Past Interglacials Working Group of PAGES, 2016, Fig. 5d therein) and scientists analyzing proxy data conclude with high confidence that high latitude surface



temperature, averaged over several thousand years, was at least 2°C warmer than present during the Eemian (Masson-Delmotte et al., 2013). The annual warming signal at high latitudes is not this pronounced in the NorESM simulations. However, a strong summer warming is simulated over the Northern Hemisphere, which is particularly important for the Eemian melt season and therefore Greenland's SMB. Especially during the early Eemian (130/125 ka), a strong seasonality is simulated globally, with

extensive DJF cooling and JJA warming in general on the Northern Hemisphere landmass. In the Southern Ocean, near surface temperatures are warmer/cooler at 125/130 ka than the pre-industrial climate, respectively, the former associated with an ice free Weddell Sea in austral winter. Arctic warming/amplification is absent or not pronounced in both seasons in early Eemian. The seasonal changes of surface temperature during the late Eemian (120/115 ka) are more modest compared to the early Eemian. During DJF, high latitude cooling is simulated in both hemispheres, with enhanced warming in most of the Northern

Hemisphere subtropical land region at 115 ka. During JJA, an overall hemisphere-asymmetric cooling pattern is simulated, with especially enhanced cooling simulated in the Northern Hemisphere land region at 115 ka.

Simulated anomalies of Arctic sea ice concentrations and thicknesses during the four Eemian time slices (Fig. 9) largely reflect the changes of surface temperature in this region. During early Eemian, March sea ice extent is close to the pre-industrial distribution, with ice thickness thinner near the central Arctic and around the coast of Greenland and Canadian Archipelago,

whereas September sea ice has a smaller extent on the Pacific side of the Arctic, with even thinner sea ice across the whole Arctic, especially north of Greenland and the Canadian Archipelago (>1.5 m ice thickness reduction). During late Eemian, March sea ice extent is also similar to the pre-industrial simulation, whereas September sea ice extent is larger on the Pacific side of the Arctic. Sea ice are thicker in both seasons, especially for 115 ka, ice thickness reduction is greater than 1.5 m in the central Arctic in March and almost across the whole Arctic in September.

**5.2   Eemian Greenland climate**

The evolution of simulated Eemian Greenland mean JJA temperature is shown in Fig. 10. As temperature during the melt season strongly influences the SMB, JJA temperature is a good indicator for the evolution of the SMB. The 125 ka time slice is the warmest for both climate models. While NorESM (top panels) shows a maximum summer warming of up to 3°C in the interior, MAR anomalies (bottom panels) reach up to 5°C at 125 ka. During the two earliest and warmest Eemian time slices,

130 and 125 ka, MAR shows particular warm and localized anomalies on the eastern and northeastern coast. The locations of these anomalies overlap with MAR regions without permanent ice cover. This localized warm anomaly is absent in NorESM. The later Eemian time slices, 120 and 115 ka, are both cooler than the pre-industrial.

The evolution of the simulated Eemian precipitation relative to pre-industrial conditions is shown in Fig. 11. The warmest periods of the Eemian, 130 and 125 ka, show more precipitation in the northwest. Especially MAR shows a positive anomaly

of up to 50% in this region in the 125 ka time slice. The coldest Eemian period, 115 ka, shows a small decrease of 10-20% in precipitation for large parts of Greenland and 120 ka shows the smallest anomalies of all the time slices. Overall NorESM and MAR show the same precipitation trends, but the MAR changes are more pronounced and show stronger regional differences which can be attributed to the higher resolution of MAR compared to NorESM.



### 5.3 Eemian Surface Mass Balance

The MAR-SEB simulation is again used as our SMB reference, also for the Eemian. The 130 ka MAR-SEB (Fig. 12e) shows a relative uniform reduction in SMB all around the Greenland margins (c.f. the MAR-SEB pre-industrial run; Fig. 7e). The strongest reduction can be seen in the southwest, where the main ablation zone is located (similar to pre-industrial). However,

there is also a noteworthy SMB reduction in the northeast. The comparison of the other four SMB models at 130 ka relative to the 130 ka MAR-SEB reference experiment is given in Fig. 12a to d. The 130 ka experiment results are shown here in detail since all model experiments show their respective lowest SMBs for this time slice, i.e. they represent our most extreme Eemian SMBs; in spite of 125 ka showing higher summer temperatures (see Fig. 10) than 130 ka. This is related to the stronger positive insolation anomaly in spring at 130 ka compared to 125 ka (not shown), giving a prolonged melt season early in the Eemian.

Under 130 ka conditions there are 60 days with a daily mean shortwave insolation above 275 $W/m^2$, in contrast to 54 days at 125 ka and only 19 for pre-industrial conditions (calculated between 58 and 70°N in the MAR domain). In regions between 40 and 70°N an insolation threshold of 275 $W/m^2$ can be used as an indicator for temperatures close to the freezing point (Huybers, 2006).

   The NorESM-forced SMB models, NorESM-PDD and NorESM-BESSI (Fig. 12a and 12c), show a more positive SMB

anomaly at the southern tip of Greenland, which is in contrast to all other model experiments. This NorESM-specific feature corresponds to a less negative SMB in the ablation zone at the margins and a more positive SMB in the interior accumulation zone relative to MAR-SEB. The coarser resolution of NorESM causes accumulation to be smeared out over the whole southern domain, instead of being localized to the southeast margin, where the highest accumulation rates are reached in the higher resolution MAR-forced experiments. Due to this resolution effect, also the total integrated SMB of the NorESM-forced exper-

iments is higher than the MAR-forced experiments. However, NorESM-BESSI (Fig. 12c) shows a lower SMB in the northeast than MAR-SEB, which causes its total SMB to be less positive than NorESM-PDD.

   From the MAR-forced experiments, MAR-PDD (Fig. 12b) shows a similar spatial SMB pattern as NorESM-PDD. However, MAR-PDD shows more ablation in the north than NorESM-PDD and there is also no resolution-related accumulation surplus in the south. The higher integrated SMB compared to the MAR-SEB reference experiments is therefore mostly related to less

ablation. Since MAR-SEB and MAR-PDD are forced with the same temperature and precipitation fields the missing ablation of MAR-PDD is caused by not accounting for insolation in the PDD model. MAR-BESSI (panel d) shows a lower SMB further inland including large areas in the north. This is a feature of both BESSI experiments, but less pronounced in NorESM-BESSI. In terms of total integrated SMB, MAR-BESSI fits MAR-SEB best, but the spatial SMB pattern is different. MAR-BESSI has less ablation around the margins, but the lower ablation is more than compensated by stronger melt in the north resulting in a

SMB more than 100 Gt/yr lower compared to MAR-SEB.

   The integrated SMB components over Greenland (including grid cells with more than 50% permanent ice cover in MAR) are shown in Fig. 12f. The accumulation (green bars) remains relatively unchanged compared to pre-industrial conditions (Fig. 7f), both the total amount and the difference between the individual SMB model experiments. The NorESM-forced experiments are slightly higher, while MAR-PDD shows the lowest accumulation. These differences are due to the different snow/rain



threshold temperature in PDD and BESSI, which are necessary due to different SMB model time steps. NorESM-PDD is less affected because the NorESM temperature is lower in all time slices compared to MAR (Fig. 10). The melt (red bars) is larger by a factor of 2 for all model experiments compared to their respective pre-industrial runs. MAR-SEB shows the highest melt, followed by MAR-BESSI. The three other model experiments show much less melt.

Note that the amount of refreeze is doubled for most model experiments compared to pre-industrial conditions. MAR-SEB shows the largest amount of refreeze, the BESSI model experiments follow with around 1/3 of the refreeze amount of MAR-SEB refreeze. The PDD models show around 1/6 of the MAR-SEB refreeze. Interestingly, NorESM-BESSI estimates almost as much refreeze as MAR-BESSI, in contrast to pre-industrial conditions where NorESM-BESSI refreeze is only half of MAR-BESSI refreeze. While the 130 ka SMB is negative for MAR-BESSI and MAR-SEB, the SMB in the other model experiments

are positive, even under the warmer conditions with reduced precipitation. Again, the NorESM-forced experiments are most positive, related to the their coarse representation of accumulation and the fact that the common ice mask cuts off parts of the NorESM ablation regions (see discussion in Sec. 4.2).

     Finally, an overview of the Greenland-integrated SMB components for all model setups and time slices is shown in Fig. 13. The accumulation (Fig. 13a) shows a slight increase in warmer periods (130 ka, 125 ka) for all model experiments. There is

a clear distinction between experiments using the PDD and BESSI models: the PDD models show lower values than their respective BESSI models (NorESM-PDD vs. NorESM-PDD and MAR-PDD vs. MAR-BESSI). This is related to the different temporal forcing (PDD: monthly; BESSI: daily) and different snow/rain temperature thresholds (see Sec. 2.1). The relatively high values for the NorESM-models can be explained by the lower resolution of NorESM, which can not resolve the relatively narrow precipitation bands at the margins of the Greenland ice sheet. Instead, the precipitation is distributed over larger regions.

Melt (Fig. 13b) and runoff (Fig. 13c) are highest in the early, warm Eemian. 130 ka shows more melt and runoff than 125 ka which is related to the prolonged melt season at 130 ka, discussed earlier in this section. The refreeze (Fig. 13d), which is basically the difference between melt and runoff, is much higher in MAR-SEB than in all other model experiments: approximately 1/3 of the melt water refreezes during the early, warm Eemian time slices; and around 1/2 refreezes during the following colder periods. The other model experiments estimate only a fraction of this refreeze. However, the experiments

using BESSI show slightly higher values than experiments using PDD. The spatial pattern of refreeze (not shown) is similar between MAR-SEB and MAR-BESSI during colder times slices (120, 115 ka, and pre-industrial), but very different during the warmer Eemian time slices (130 and 125 ka). The MAR-SEB refreeze pattern remains similar for all time slices, with an intensification in the warmer Eemian time slices all around the margins of Greenland, particularly in the south. In contrast, most MAR-BESSI refreeze during the two warm time slices occurs along the southeastern and northeastern margins. The

MAR-BESSI experiments in general show smaller refreeze quantities, while also increasing in the two warm time slices.

     The total SMB (panel e) shows a clear difference between NorESM- and MAR-forced models. NorESM based models are offset towards positive values, due to higher accumulation and less melt. The SMB of the MAR-BESSI experiment is consistent with MAR-SEB for all time slices. MAR-PDD is consistent with MAR-SEB for the cold time slices (120, 115, and pre-industrial), but does not capture the same negative SMBs at 130 and 125 ka.





These results, particularly the differences in melt during the warmer Eemian time slices indicate two things. Firstly, it is challenging not to include shortwave radiation in a SMB model when investigating the Eemian, because the melt might be underestimated. Secondly, a SMB model can not fix shortcomings of a global climate forcing (i.e., low resolution like here, or other deficiencies). Both, the climate as well as the type of SMB, are important for an accurate simulation of Greenland's

SMB, while either of the two can be more important depending on the climate state and particularly the prevailing insolation pattern.

## 6   Discussion

The Eemian interglacial is characterized by a positive Northern Hemisphere summer insolation anomaly giving warmer summers over Greenland. This challenges the Eemian SMB estimates based on PDD as insolation changes are not implicitly

included. Here, we assess how the resolution of the climate forcing and the choice of SMB model influences Eemian Greenland SMB estimates. A relatively high resolution Eemian global climate simulation (with NorESM) is combined with regional dynamical downscaling (with MAR). Previous studies, employing downscaled SMB over Greenland, use either low complexity models (Robinson et al., 2011; Calov et al., 2015), or forcing fields from low resolution global climate models (Helsen et al., 2013) as input. Unfortunately, the uncertainties associated with the global climate simulation add a major constraint to any high

resolution Greenland SMB estimate. For example, Eemian global climate model spread has been hypothesized to be related to differences in the simulated Eemian sea ice cover (Merz et al., 2016). Furthermore, sensitivity experiments with global climate models by Merz et al. (2016) show that sea ice cover in the Nordic Seas is crucial for Greenland temperatures (i.e., a substantial reduction in sea ice cover is necessary to simulate warmer Eemian Greenland temperatures in agreement with ice core proxy data). However, the quantification of Eemian global climate simulation uncertainties is beyond the scope of this paper and we

refer the reader to Earth System Model intercomparisons focusing on the Eemian (Lunt et al., 2013; Bakker et al., 2013), as well as studies seeking to merge data and models (e.g., Buizert et al., 2018), for details on efforts to improve Eemian climate estimates and reduce global climate uncertainties.

The Eemian climate simulations, with NorESM and MAR, use the present-day topography of Greenland, neglecting any topography change, or freshwater forcing from a melting ice sheet. Given the lack of a reliable Eemian Greenland topography

or meltwater estimate, this is a shortcoming we choose to accept. Merz et al. (2014a) discuss global climate simulations using various reduced Eemian Greenland topographies without finding any major changes of the large-scale climate pattern. However, there is a clear impact of Greenland topography changes on the local surface air temperature, given that the surface energy balance is strongly dependent on the local topography (e.g., due to changes in local wind patterns and surface albedo changes as a region becomes deglaciated). The same is true for the relationship between Greenland topography and Eemian

precipitation patterns Merz et al. (2014b) — large-scale patterns are fairly independent of the topography, but local, orographic precipitation follows the slopes of the ice sheet. The impact of orographic precipitation is also clear when transitioning from low to high resolution in models: as an example, for the pre-industrial simulation with MAR in Fig. 11 the higher resolution



topography results in enhanced precipitation along the better resolved sloping margins of the Greenland ice sheet (e.g., note the southeast margin).

At 130 ka the Greenland ice sheet was likely larger than today, as the climate was transitioning from a glacial to an inter-glacial state. A smaller sized ice sheet leads to higher simulated temperatures over Greenland due to the lower altitude of the

surface and the albedo feedback in non-glaciated regions. Additionally, neglecting the meltwater influx to the ocean from the retreating glacial ice gives warmer simulated air temperatures. As a result, the simulated 130 ka temperatures are assumed to be warmer than they were in the past, causing a low simulated SMB. Similarly, the present-day ice sheet, and particularly the ice mask, is likely misrepresenting the 125 ka state of Greenland. A larger ice sheet will include regions of potentially highly negative SMB, lowering the integrated SMB, i.e., the simulated integrated 125 ka SMBs are likely also too low to be realistic.

Basically, only a fully coupled ice sheet-atmosphere-ocean simulation would be able to realistically account for evolving ice sheet configuration and meltwater input to the ocean. Here, the simulated 130 ka SMB is discussed in more detail, not because it is assumed to be the most realistic, but because it provides the most extreme SMB cases within our Eemian climate simu-lations. Furthermore, the spatial SMB pattern does not change significantly between 130 and 125 ka in our simulations, i.e., conclusions drawn for 130 ka are also true for 125 ka.

The comparison of different SMB models requires a common ice sheet mask which is always a compromise. Vernon et al. (2013) show that approximately a third of the intermodel SMB variation between four different regional climate models is due to ice mask variations at low altitude (models forced with 1960-2008 reanalysis data over Greenland). Resolution-dependent ice sheet mask differences between NorESM- and MAR-derived SMBs are important here. Due to the larger NorESM grid cells, the NorESM ice mask extends beyond the common MAR ice mask (not shown), and as a result the NorESM ablation

zone is partly cut off when using the common MAR ice mask. As a consequence there is less ablation in the NorESM-forced SMB model experiments than in the MAR-forced experiments. The direct comparison between NorESM- and MAR-derived SMBs is therefore challenging. However, the PDD and BESSI models are run with both climate forcing resolutions to allow a consistent comparison, independent of the ice mask.

For the "cooler climate states", similar to pre-industrial (i.e., 120 and 115 ka), the different resolution of the climate forcing

shows the largest influences on the derived SMBs. The complexity and physics of the SMB model is of secondary importance during these periods. This comes as no surprise, as the PDD parameters employed are based on modern observations, and the intermediate model, BESSI, was tuned to represent MAR-SEB under pre-industrial conditions. As discussed earlier, the resolution-dependent difference is caused by higher accumulation in the south, but also less ablation due to the differences in ice sheet mask.

In the "warmer climate states" (i.e., 130 and 125 ka) the complexity of the SMB model becomes the dominant factor for the derived SMBs. SMB model experiments forced with the high resolution climate, and a representation of solar insolation, show spatially integrated Eemian SMBs which are negative. Testing the SMB models with two different climate forcing resolutions as input illustrates that it is essential to resolve local climate features — an inaccurate climate (e.g., due to coarse topography) will result in an inaccurate SMB. Besides coarse representation of Greenland's topography, changes in ice sheet topography



and sea ice cover are likely to have a major impact on the climate over Greenland during the Eemian. However, as mentioned above, it is beyond the scope of this study to evaluate the uncertainty in the simulated Eemian climate forcing.

The biggest differences between SMB components, both between the individual models and between the climate time slices, arise for melt and refreeze (runoff basically represents the difference of these two). The PDD-derived experiments lack melting compared to the other experiments, due to neglecting insolation. MAR-PDD shows slightly more melt partly because it uses the higher resolution climate (i.e., a climate derived with better representation of surface processes and surface albedo). NorESM-PDD and MAR-PDD show the least refreeze, related to the simple refreeze parameterization, but also due to the smaller amount of melt. The intermediate model, BESSI, shows more melt in the warm Eemian, and almost matches the values of the MAR-SEB reference experiments if forced with the regional climate. However, BESSI can not compensate for the shortcomings of the lower resolution climate in NorESM-BESSI. In general, BESSI shows slightly more refreeze than PDD, but refreeze remains underrepresented compared to MAR-SEB. This is likely related to a fairly crude representation of the changing albedo (i.e., albedo is changed with a step function from dry to wet snow to glacier ice — a more accurate albedo representation is in development). BESSI also does not have a daily cycle, e.g., neglecting colder temperatures at night where refreeze might occur. Furthermore, BESSI shows wide regions where the complete snow cover is melting away, exposing glacier ice and prohibiting any further refreeze in these regions (particularly under warm Eemian conditions). As a result, the shift in albedo causes more melting in these regions (e.g., areas with negative SMB anomaly in the 130 ka MAR-BESSI experiment in Fig 12c).

MAR-SEB stands out with the highest values of melt and refreeze. Particularly, the refreeze is much larger as in all other experiments. During cooler time slices (120, 115 ka, pre-industrial) MAR-SEB shows twice the refreeze amount as MAR-BESSI. During warmer times slices (130, 125 ka) the ratio goes up to at least triple the amount. This can partly be explained by the fact that MAR uses a higher temporal resolution, i.e., MAR is forced with 6-hourly NorESM climate and runs with a model time step of 180 seconds. BESSI on the other hand uses daily time steps to calculate its SMB. The lower temporal resolution of the BESSI forcing causes a smoothing of extreme temperatures resulting in less melt and refreeze. Tests forcing BESSI with a daily climatology instead of a daily transient, annually varying climate, show less refreeze for similar smoothing reasons (not shown). During the cooler periods, MAR-SEB produces more melt and refreeze than the other model experiments. This occurs all around the margins of Greenland, similar as in the MAR-BESSI experiment (but lower values in MAR-BESSI). Under the warmer Eemian conditions, MAR-SEB simulates a refreeze intensification in the same regions, with particularly strong refreeze in the south. In contrast, MAR-BESSI shows most refreeze in the southeast and the northwest.

Comparing the differences between SMB models under pre-industrial (Fig. 7) and Eemian conditions (Fig. 12) indicates that the inclusion of solar insolation in the calculations of Eemian SMB is important. If this were not the case, the differences between the individual SMB experiments would be more similar for pre-industrial and the Eemian conditions, and the two latter figures would look more similar. However, any model that accounts for solar insolation strongly relies on a correct representation of the atmosphere (e.g., the most sensitive parameter of BESSI is the emissivity of the atmosphere, Tab. 2). This high dependency on a correct atmospheric representation (e.g., cloud cover) is also true for a full surface energy model like in MAR-SEB. It is essential to keep this in mind when evaluating simple and more advanced SMB models for paleo applications. The PDD approach for example has been used extensively to calculate paleo SMBs, but it also has been criticized



often. However, the most reliable paleoclimate proxies are air temperature and precipitation, and it is hard to argue why a energy balance model which needs poorly constrained information (e.g., net radiation) would produce more reliable results for paleo ablation than a simple PDD model (Fausto et al., 2009).

The comparison of previous Eemian studies in Sec. 3 shows the importance of the climate forcing for estimating the ice sheet
extent and sea level rise contribution. Most studies used a combination of the positive degree day method and proxy-derived or global model climate and the estimated ice sheets differ strongly in shape. All studies use similar ice dynamics approximations. Therefore, it is a fair assumption that the differences are a result of the climate used. The more recent studies with further developed climate and SMB forcing, also lack a coherent picture. But since they use different climate downscaling and different SMB models, it is hard to separate the influence of climate and SMB model. The present study reveals strong differ-
ences between SMB model types particularly during the warm, early Eemian. However, it remains challenging to quantify the uncertainty contributions related to global climate forcing (not tested here) and to SMB model choice. More sophisticated SMB models might seem like a obvious choice for future studies of the Eemian Greenland ice sheet due to their advanced representation of atmospheric and surface processes. However, as long as the uncertainty of Eemian global climate simulations can not be narrowed down further (e.g. cloud cover and other poorly constrained atmospheric variables which influence the surface
energy balance) different SMB models should be included in Eemian ice sheet simulations to capture uncertainties related to model selection in paleo applications. Since it is not feasible to perform transient fully-coupled climate-ice sheet model runs with several regional climate models, it is desirable to perform Eemian ice sheet simulations within a model intercomparison covering a range of different (high resolution) climate forcings and a range of SMB models to capture uncertainties in the best possible way. Furthermore, if lower resolution global climate is used, it might be worth to investigate options for correcting
not just the temperature, but also the precipitation/accumulation fields.

## 7 Conclusions

In this study a relatively high resolution global climate model (NorESM) and a regional climate model (MAR), constrained by the global climate fields, are used to estimate the surface mass balance (SMB) during the Eemian interglacial employing three types of SMB models — a simple positive degree day (PDD) model, an intermediate complexity model (BESSI), and a full
surface energy balance model (implemented in MAR). The Eemian interglacial is characterized by a warm summer climate caused by a positive Northern Hemisphere summer insolation anomaly which renders insolation representation in SMB models important. While all SMB models work similarly well during cooler climate conditions (120, 115 ka and pre-industrial), forcing the various SMB models with the two climate resolutions reveals the importance of representing regional climate features, like the narrow southeastern precipitation band typical for Greenland. During the warm early Eemian the SMB model choice
becomes very important, due to different representation of insolation in the models. The full surface energy balance model forced with the regional climate exhibits the largest values for melt and refreeze compared to all other experiments in the present model pool. Despite, the most sophisticated representation of surface processes and topography in this study, the results is also dependent on the global climate simulations. While the individual SMB components are very different between

SMB models we recognize that a further improved intermediate complexity SMB model (i.e. albedo parameterization) would be very useful for forcing ice sheet models on paleo time scales. If the overall SMB pattern is simulated correctly without using full energy balance models, then ice sheet models will produce similar results, since the individual components (e.g. meltwater and refreeze) are only used to a limited degree by state-of-the-art paleo ice sheet models. In conclusion, both the climate as well as the type of SMB model are important for an accurate simulation of Greenland's SMB. Which of the two becomes most important is dependent on the climate state and particularly the prevailing insolation pattern. To improve the Eemian SMB estimate, further effort needs to be put in developing fully-coupled regional climate-ice sheet models and making them efficient enough to be run over whole glacial-interglacial cycles. We deem Eemian climate model intercomparions combining with various SMB models to be the best way to evaluate and ultimately lower Eemian SMB uncertainties.

## 8    Code availability

The NorESM model code can be obtained upon request. Instructions on how to obtain a copy are given at: https://wiki.met.no/noresm/gitbestpractice. The PDD python script is available at: https://github.com/juseg/pypdd. BESSI is under active development. For more information contact Andreas Born (andreas.born@uib.no). The MAR code is available at: http://mar.cnrs.fr.

## 9    Data availability

The full set of NorESM model data will be made publicly available through the Norwegian Research Data Archive at: https://archive.norstore.no upon publication. The MAR, BESSI, PDD experiment simulations are available upon request from the corresponding author.

*Author contributions.*    AP and KHN designed the study with contributions from AB and PML. CG and KHN provided the NorESM simulations. SLC performed the MAR simulations. MI and AB wrote the BESSI code with contributions from TS. AP made the figures and wrote the text with input from KHN, AB, PML, CG (Sec. 2.1 and 5.1), and SLC (Sec. 2.1). All authors commented on the final version of the manuscript.

*Competing interests.*    The authors declare that they have no conflict of interest.

*Acknowledgements.*    The research leading to these results has received funding from the European Research Council under the European Community's Seventh Framework Programme (FP7/2007-2013) / ERC grant agreement 610055 as part of the ice2ice project. We would like to thank all authors cited in Fig. 3 and 4 for sharing their Eemian Greenland ice sheet results. We would like to thank Anne-Katrine Faber for providing the shallow ice core data she compiled during her PhD. Furthermore, we would like to thank Tobias Zolles for discussions of the BESSI results. PML's contribution was supported by the RISES project of the Centre for Climate Dynamics at the Bjerknes Centre for Climate Research.



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





**Figure 5.** Annual mean near surface temperature and precipitation simulated with NorESM and MAR for pre-industrial conditions. The NorESM temperature is corrected with the temporally and spatially *3D lapse rate* (see Sec. 4.2). The top row shows modeled temperatures with observations from ice cores and weather stations plotted on top. Additionally, scatter plots of observed vs. modeled for each model are presented. The bold gray lines represent the 1:1 diagonal and hence a perfect fit between model and observations. The bottom row shows the same for annual mean precipitation.





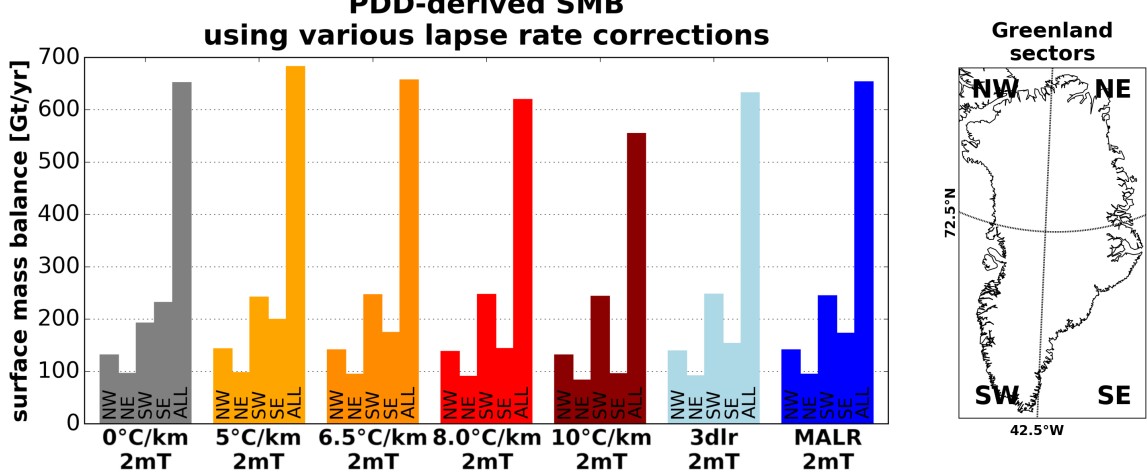

**Figure 6.** Sensitivity of the PDD-derived SMB to the applied temperature lapse rate correction (to low resolution climate). The bars show the integrated SMB over the GrIS, and its regional contributions. *0°C/km* refers to the uncorrected temperature, *5 to 10°C/km* represent spatially uniform temperature lapse rates, *3dlr* is the 3D lapse rate derived from the vertical NorESM temperature column, and *MALR* is the moist adiabatic lapse rate calculated from the thermodynamic state of the NorESM surface air layer.





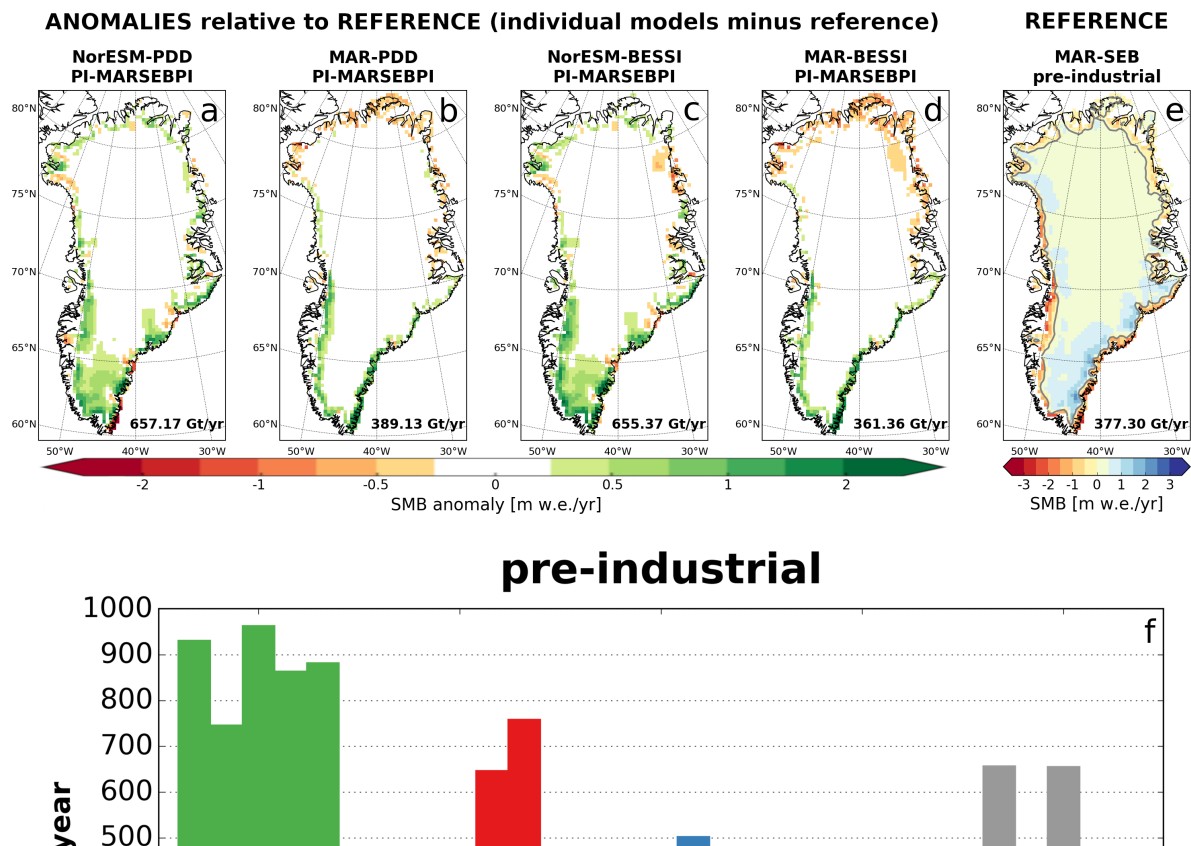

**Figure 7.** Comparison of the simulated **pre-industrial SMB** from all five model setups. The first row shows the spatial map of SMB for NorESM-PDD, MAR-PDD, NorESM-BESSI, MAR-BESSI, and MAR-SEB, respectively. Our reference, MAR-SEB (panel e), is shown in absolute values, while the four simpler models (panel a-d) are shown as anomalies to MAR-SEB. The total SMB integrated over all of Greenland (including grid cells with more than 50% permanent ice) is given in numbers on each panel. The same ice mask is used for the bar plots in panel f. The bar plots show the individual components contributing to the total SMB (in Gt/year) for each model.





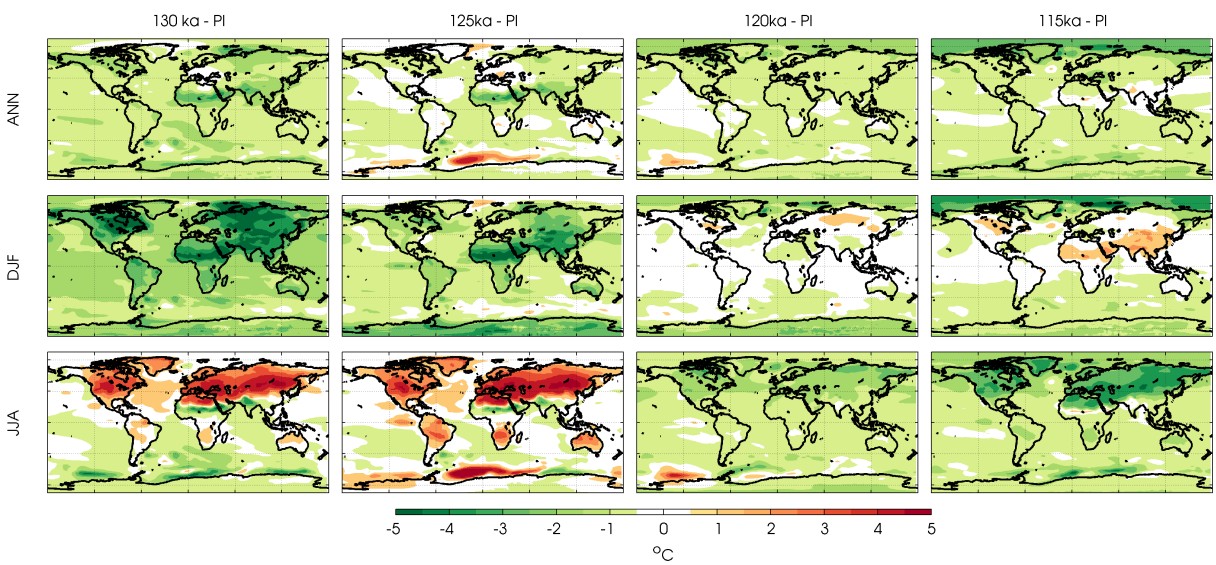

**Figure 8.** Simulated changes of near surface temperature for the Eemian experiments relative to the pre-industrial experiment (PI). The top row shows the annual mean, and the middle and lower rows show the DJF and JJA mean, respectively. The columns show the temperature changes for the 130, 125, 120, 115 ka from left to right. Model results are annual means over the last 100 years of model integration. A Latitude/Longitude grid is indicated with dashed lines with a 60° spacing.



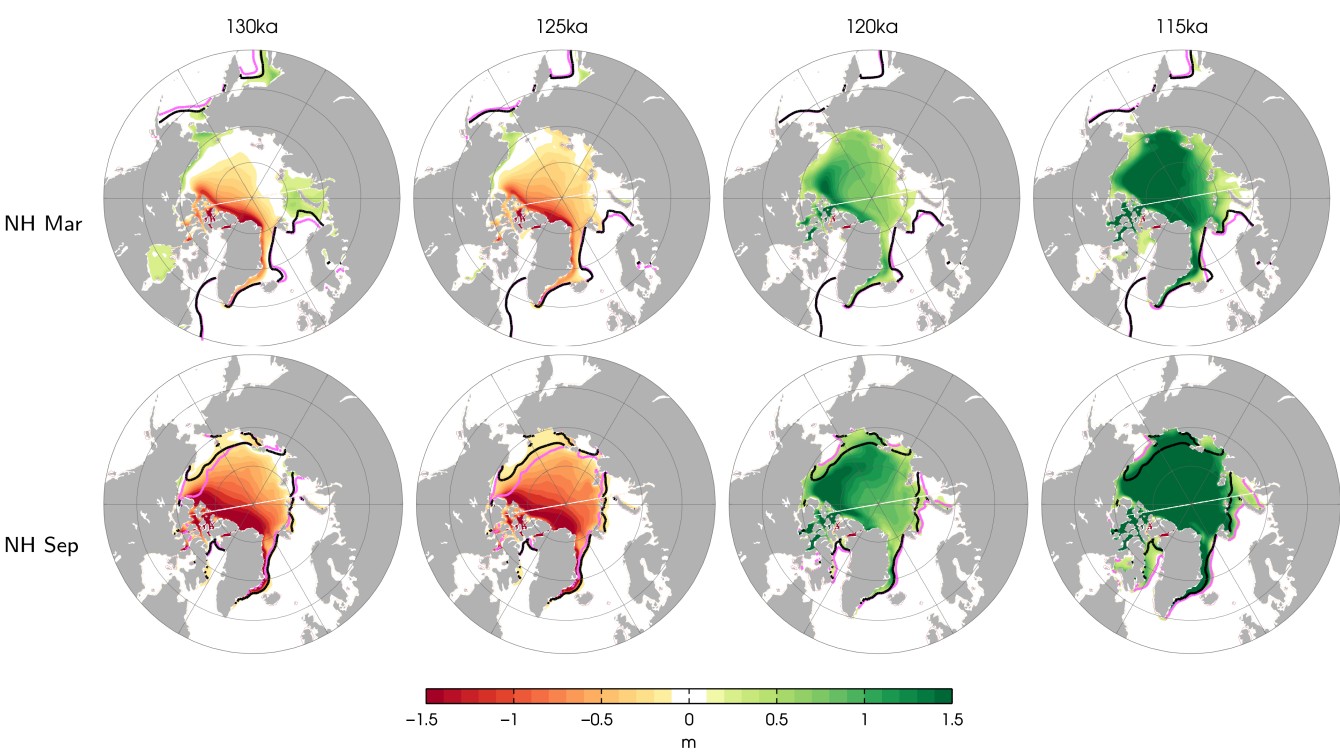

**Figure 9.** Simulated changes of Arctic sea ice thickness for the four Eemian experiments relative to the pre-industrial experiment. The top and bottom row show the sea ice changes in March and September, respectively. The left to right columns show the sea ice changes for the 130, 125, 120, 115 ka experiments, respectively. The solid black and magenta contour lines show the 15% sea ice concentration for each Eemian experiment and the pre-industrial experiment, respectively. Model results are annual means from the last 100 years of model integration. A Latitude/Longitude grid is indicated with gray lines with a 10/60° spacing.





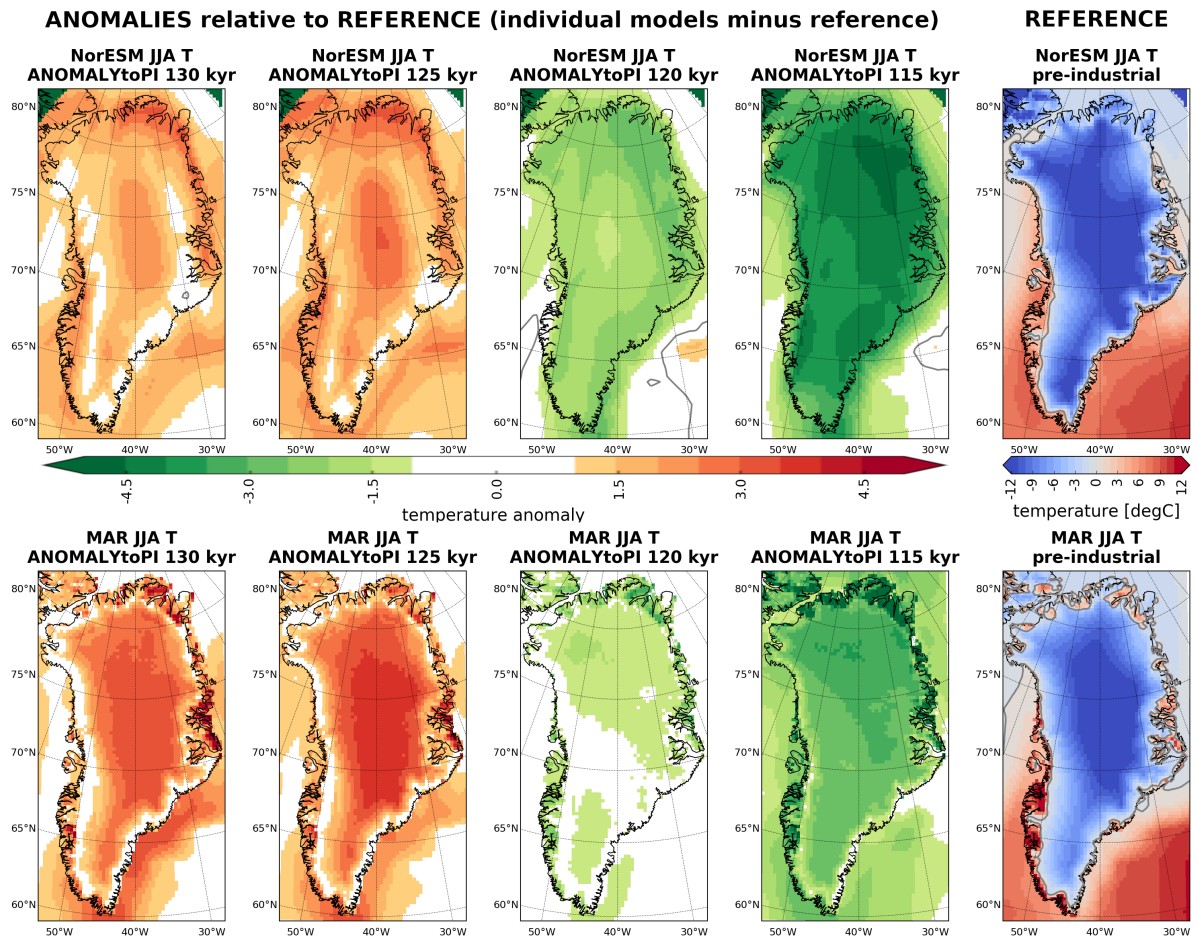

**Figure 10.** Evolution of the simulated **summer temperature (June-July-August; JJA)** during the Eemian Interglacial. NorESM results for Greenland are shown in the top (the temperature is lapse rate corrected; see Sec. 2.2) and MAR in the bottom row. The Eemian temperatures are shown as anomalies relative to the pre-industrial simulation. The solid gray line indicates the 0°C isotherm.



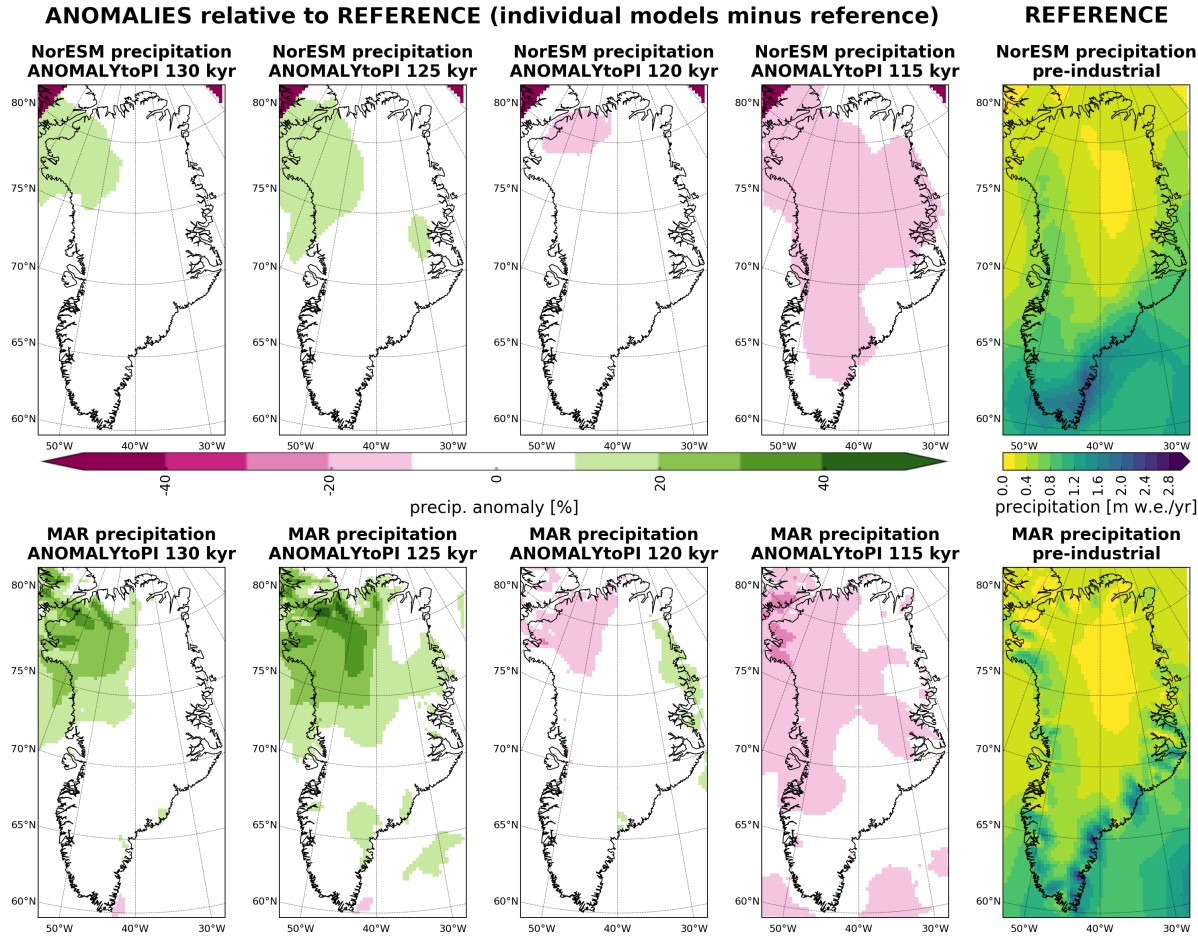

**Figure 11.** Evolution of the simulated **annual precipitation** during the Eemian Interglacial. NorESM is shown in the top and MAR in the bottom row. Eemian time slices are shown as anomalies relative to the pre-industrial simulation.



**Figure 12.** Comparison of the simulated **130 ka SMB relative to the 130 ka MAR-SEB** with all five model combinations. The first row shows the spatial map of SMB for NorESM-PDD (panel a), MAR-PDD (b), NorESM-BESSI (c), MAR-BESSI (d), and MAR-SEB (e), respectively. Our reference, MAR-SEB, is shown in absolute values, while the four simpler models (panel a-d) are shown as anomalies to MAR-SEB. The total SMB is integrated over all of Greenland (including grid cells with more than 50% permanent ice). The same mask is used for the bar plots in panel f. The bar plots show the absolute values for each component of the SMB for the same experiments.





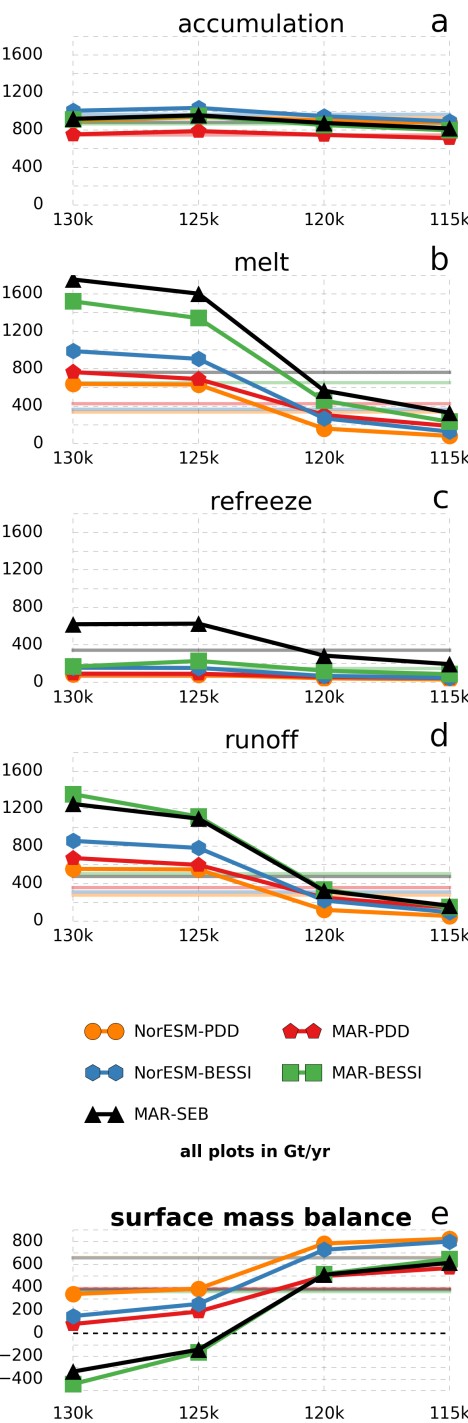

**Figure 13.** Eemian evolution of the SMB components (integrated over grid cells with more than 50% ice cover in MAR; Gt/yr). Pre-industrial values for each model are shown as shaded lines.