# Peer review of "Eemian Greenland SMB strongly sensitive to model choice"

_Climate of the Past, 2018_

## Referee Comment (RC1) · A. Robinson (Referee) · 8 Aug 2018

This study is focused on understanding the challenges and sources of uncertainty of simulating the surface mass balance (SMB) of the Eemian interglacial period. Steady-state time slice simulations are performed for the Eemian and the present day, with global and regionally downscaled climatic forcing applied to several combinations of SMB models. The manuscript does a good job of describing many aspects of Eemian smb modeling that are often overlooked (seasonal changes in climate, sea-ice extent, lapse rate validation). The review of past Eemian sea-level contribution estimates is also well done, even if it is only part of the motivation for the current work rather than the main focus. I think that the paper should be published after minor revisions, explained below.

This is an excellent time-slice study with a good experimental design and thorough analysis. However, it is missing any insight into the role of feedbacks in transient coupled experiments where the ice-sheet topography could evolve. This could arguably be as important as the inherent bias that a particular smb model imposes, or even more so – see Robinson and Goelzer (2014), for example. I suggest adding some discussion of this point (note that this is a different point than that of the last paragraph on Page 18, and the first paragraph of Page 19 is more focused on whether a given time slice is realistic).

While I found the analysis very thorough, it was difficult to agree with the overall conclusions reached by the authors. For example, I disagree with this sentence from the abstract: "We suggest that future Eemian climate model inter-comparison studies are combined with different SMB models to quantify Eemian SMB uncertainty estimates." To me this is a strange conclusion to make, or perhaps I don't understand the phrasing clearly. Should we believe PDD is providing added information to an energy balance model? This also comes up in the last paragraph of the Discussion. The authors seem to conclude that all SMB models are needed, because emissivity of the atmosphere is uncertain. This is a strong conclusion, but here nothing was done with emissivity. Further, wouldn't a more prudent conclusion be that deeply uncertain parameters in complex models should include sensitivity experiments (parameter perturbation) rather than simply reverting to simpler models known to lack important processes?

Along those lines, I think it would have been quite interesting to see if using different parameter values (for example changing the emissivity of the atmosphere), it would be possible to bracket the MAR-SEB results on both sides with MAR-BESSI (SMB at 130ka showing negative and positive anomalies). That would go a long way towards showing that lower complexity smb models can be useful, but several simulations may be necessary to sample the uncertainty. [This is only a suggestion, not a requirement for publication.]

Generally, the manuscript could use a revision for English usage as well. Some mis-

takes are highlighted below. Particularly, I noticed the article "the" missing in many instances.

== Minor comments =====

Page 1, line 2: Eemian interglacial => Eemian interglacial period

Page 1, line 5: "introduces uncertainties" sounds a bit strange, consider rephrasing.

Page 1, line 10: the calculation of insolation should be straightforward – do you mean shortwave radiation at the surface?

Page 1, line 12: simulated climate => simulated climate,

Page 2, line 1: Past interglacials => Past interglacial periods [Generally this should be changed throughout, as "interglacial" is only and adjective.]

Page 2, line 7: pre-industrial => pre-industrial period

Page 4, line 30: surface air temperature => near-surface air temperature [?]

Page 4, line 34: "The only process it neglects" <= This is a strong statement, consider rephrasing.

Page 5, table 1: Units of PDD factors should be "mm/K/day"

Page 6, line 15: linearly => bilinearly [?]

Page 6, line 7: This 30 years => These 30 years

Page 8, Fig. 1: Lighter colors in the lower boxes would make this figure easier to read.

Page 10, line 13: "with an adapted PDD scheme" <= the ITM equation used by Robinson et al. (2011) and Calov et al. (2015) is not a PDD scheme, it is a "linearized energy-balance" scheme (originally published by Pollard, 1980).

Page 13, line 35: are we using => we use

Page 14, line 20: refreeze => refreezing [Change everywhere it appears as a noun.]

Page 15, line 6: "warmer/cooler at 125/130 ka" <= Consider reversing the time order here for consistency with elsewhere.

Page 15, line 7: I think Arctic warming and amplification are not synonymous, consider revising here somewhat for clarity.

Page 15, line 13: During early Eemian => During the early Eemian

Page 15, line 18: Sea ice are => Sea ice is

Page 18, first paragraph: This seems more like Discussion than Results.

== References =====

Robinson, A. and Goelzer, H.: The importance of insolation changes for paleo ice sheet modeling, The Cryosphere, 8, 1419-1428, https://doi.org/10.5194/tc-8-1419-2014, 2014.

---

## Referee Comment (RC2) · Anonymous Referee #2 · 18 Aug 2018

General Comments

This paper considers sources of uncertainty in simulating Greenland ice sheet surface mass balance (SMB) during the Eemian interglacial. The authors use a global Earth System model (NorESM), a regional climate model (MAR), and three kinds of SMB model (a positive-degree-day scheme, a model of intermediate complexity, and a full surface-energy-balance model) to assess the sensitivity of Eemian SMB to climate model resolution and SMB model complexity. The authors find that for earlier Eemian time slices (130 and 125 ka, with high summertime Northern Hemisphere insolation), results are sensitive to model choices, with regionally-forced SMB models giving a more negative SMB than globally-forced models, and with the PDD model underestimating melting compared to the more complex models. For later Eemian time

slices (120 and 115 ka, with lower insolation), the SMB model is less critical, but SMB remains sensitive to the resolution of the forcing climate model.

The study is well designed, using a novel combination of models to draw useful inferences about SMB sensitivity for the Eemian. The authors give a broad review of earlier work and clearly describe their experimental methods. The analysis is clear and detailed, and the conclusions (with exceptions noted below) are generally well supported by the text and figures.

My main concern is that some of the conclusions are not well supported by the simulation results. I would suggest rewriting or removing some of these statements, as described below. Also, the text would benefit from some editing for English grammar; see Technical Corrections. Otherwise, the authors provide a solid and useful analysis of Eemian SMB sensitivity, and I suggest publication with minor revisions.

Specific Comments

I suggest a modified title. The current title emphasizes the sensitivity of the Eemian SMB to SMB model choice, whereas the text suggests an equally important role for the kind of climate forcing (high-resolution RCM v. lower-resolution GCM).

p. 1, l. 14: "We suggest that future Eemian climate model inter-comparison studies are combined with different SMB models to quantify Eemian SMB uncertainty estimates." Unless I misunderstand how "SMB model" is defined, this statement is not well supported. The text identifies three kinds of SMB model: PDD, intermediate complexity (BESSI), and full surface-energy-balance (as in MAR). The results suggest that PDD schemes are inappropriate for the early Eemian, when insolation differed markedly from present-day. While BESSI results are closer to MAR, I don't see an argument that BESSI results are in any way more accurate or credible than MAR results. I would infer that future studies should use MAR-SEB or a comparable scheme, in order to minimize uncertainties. More generally, one should always use the most realistic, best validated model that is computationally practical, unless it can be shown that running a simpler,

cheaper model yields closely similar results.

There are many other sources of uncertainty for simulated Eemian SMB, notably the absence in this study (as the authors point out) of time-varying topography. It seems more fruitful for future studies to explore other sources of uncertainty rather than revisit simple SMB models.

p. 2 l. 35: "the amplification of summer warming over Greenland has been found to be effective". I'm not sure what is meant; effective for what?

p. 3, l. 2: Overall, I found Section 2 to be a very clear and helpful description of the models and methods.

p. 4, l. 33: "The only process it neglects. . ." I suggest "It neglects. . .", since there are bound to be other neglected processes.

p. 6, ll. 24ff: When I read this the first time, I wondered whether the study used the same static surface topography for each time slice. It does, as stated later, but I suggest stating it here.

p. 9, ll. 1ff: I liked the comprehensive description of earlier studies and their limitations. However, this section might fit better into the overall structure if swapped with Section 2.

p. 9, l. 4: A broad range of 0.4 to 5.6 m is given, but the more recent studies have an upper bound of ∼3 m. Does this narrowing of the range (combined with the more recent ice core evidence) suggest that the high-end estimates likely are too high?

p. 10, l. 33: Please say what is meant by "model consistent", or otherwise give a bit more detail about how the 3D lapse rate is computed.

p. 13, l. 15: "the ablation in the SW reaches much lower values". Please clarify whether ablation is lower, or the SMB is lower (i.e., more negative).

p. 15, l. 2: Can you say why the annual warming signal is less pronounced in NorESM?

Do you suspect a winter cold bias, a summer cold bias, or both? (I don't think this is critical to explain, just helpful if you can make an educated guess. Similarly for the next question.)

p. 15, l. 7: Are you able to explain why Arctic amplification is mostly absent in the early Eemian?

p. 18, l. 2: I think "challenging" is not the right word here; not including SW for the Eemian seems like a more fundamental flaw. Maybe "highly problematic"? Similarly, "challenges" in l. 9 below could be replaced with something like "complicates".

p. 18, l. 4: I suggest removing "or other deficiencies", since deficiencies apart from coarse resolution haven't been discussed.

p. 18, l. 11: I'm not sure NorESM should be described as "relatively high resolution". Its resolution is low compared to MAR, and is not high compared to other IPCC-class ESMs. Some global ESMs, for example, run with a 1 degree rather than 2 degree atmosphere.

Section 6: The first part of the discussion appropriately focuses on big issues such as variable topography and climate forcing resolution. Later, e.g. the second full paragraph on p. 20, it gets into finer details such as refreezing and temporal resolution in BESSI, which might fit better in Section 5.

p. 18, l. 27: A discussion of evolving Greenland topography should refer to the study of Ridley et al. 2005 (in the context of future warming and deglaciation), and possibly some more recent coupled ESM-ISM studies.

p. 19, l. 5: "neglecting the meltwater influx to the ocean from the retreating glacial ice gives warmer simulated air temperatures". Can you say briefly why this is the case?

p. 21, l. 1: "it is hard to argue why a energy balance model which needs poorly constrained information (e.g., net radiation) would produce more reliable results for paleo ablation than a simple PDD model". I don't think this statement is well supported.

For example, incoming solar insolation is very well constrained by orbital calculations, and this alone is a good reason that an energy-balance model might produce more reliable paleo ablation than a PDD model.

p. 21, l. 15: "different SMB models should be included in Eemian ice sheet simulations to capture uncertainties". I disagree with this statement. It is true that there will always be some uncertainties in atmospheric variables (such as cloud cover) that influence the surface energy balance. But it does not follow that "the uncertainty of Eemian global climate simulations cannot be narrowed down further." (For instance, one could build a better cloud model.). Also, I see no reason not to use the best computationally afford-able SMB model (either MAR's SEB model or something comparable). See comments above for p. 1, l. 14.

p. 21, l. 18: "it is desirable to perform Eemian ice sheet simulations within a model intercomparison covering a range of different (high resolution) climate forcings and a range of SMB models". Please define what is meant by high resolution. E.g., finer than 1 degree? Fine enough to capture orographic precipitation and narrow ablation zones?

I'm again unclear on the value of a range of SMB models for UQ, unless the range includes other models with SEB schemes comparable to MAR (e.g., RACMO). Also, it could be valuable to explore a range of parameter settings within MAR, to the extent that certain parameters are uncertain and tunable.

p. 22, l. 1: "we recognize that a further improved intermediate complexity SMB model (i.e. albedo parameterization) would be very useful for forcing ice sheet models on paleo time scales." I agree that models like BESSI could be improved for paleo simula-tions, but I don't see why an improved intermediate model would be preferable to SMB forcing from a detailed RCM. Assuming that you're already using MAR or another RCM for dynamical downscaling, why not just use the RCM's SMB?

p. 22, l. 7: "further effort needs to be put in developing fully-coupled regional climate-ice sheet models and making them efficient enough to be run over whole glacialinterglacial cycles". I'm unclear on the role of global models here. Is the idea that the RCM would be run interactively with a global climate model, or just the ice sheet model? Also, what is meant by a whole glacial-interglacial cycle? Do you mean an interglacial time scale (∼10 kyr) or a full glacial time scale (∼100 kyr)?

I think that coupled GCM-ISMs have a role to play, which is not acknowledged here. Other GCMs/ESMs could prove to be more accurate than NorESM for Eemian SMB studies, using some combination of higher (or spatially variable) resolution, improved cloud and snow physics, and SEB schemes with subgrid elevation classes. Even if the SMB from a global ESM is less accurate than the dynamically downscaled SMB from MAR, this disadvantage could be offset by the benefits of simulating topographic feedbacks in a global model.

p. 22, l. 8: I disagree with the last sentence of the conclusions (in particular, "combining with various SMB models"), for the reasons stated above.

Technical Corrections

p.2 l. 10: "While" -> "However"

p. 2 l. 17: "Global Circulation Models" -> "Global Climate Models"

p. 2, l. 19: No caps in Surface Mass Balance. Likewise Surface Energy Balance, l. 24

p. 2, l. 27: Delete "due to"

p.2, l. 28: "which is the reason for" -> "which are primarily responsible for"

p. 4, l. 27: typo, "Ber/ge/n"

p. 5, l. 2: "Firn densification is realized with models...". Awkward phrasing; please reword.

p. 7, l. 6: "This 30 years" -> "These 30 years of output..."

p. 7, l. 8: "downwards "-> "downward"

p. 8, l. 14: Add comma after "topography"

p. 9, l. 14: Delete "a" before "Eemian"

p. 10, l. 4: "we are not discussing the ice dynamics used further." Suggest "we do not further discuss the ice dynamics."

p. 11, Fig. 3: "Nisancioglu" is misspelled. Suggest adding "Simulated" before "sea level rise" in the title.

p. 12, l. 9: Add units after "5"

p. 13, l. 6: No commas needed in this sentence.

p. 13, l. 22: No comma after "Both"

p 13, l. 29: "lower-resolution" (with a hyphen)

p. 13, l. 35: "are we using" -> "we are using"

p. 14, l. 9: can not -> cannot

p. 15, l. 14: "with ice thickness thinner" -> "with ice thinner"

p. 15, l. 18: "is thicker" -> "are thicker". Also, do you mean an ice thickness increase?

p. 17, l. 11: Delete "the" before "their"

p. 18, l. 4: Suggest "Both the climate and the type of SMB are important"

p. 18, l. 30: Misplaced parentheses for Merz citation

p. 19, l. 6: "assumed" is not the right word, since you've given an argument. Suggest "...130 ka temperatures are likely warmer than the actual temperatures, resulting in..."

p. 19. l. 24: No quotes needed for "cooler climate states". Likewise below for "warmer climate states".

p. 21, l. 7: "assumption" -> "inference"

p. 21, l. 32: No comma needed after "Despite"

p. 29, Fig. 5: Use the same symbol for, e.g., ice cores in both temperature and precipitation plots. Should l. 2 of the caption read "temporally and spatially varying 3d lapse rate"?

p. 32, Fig. 8: The panels are small and hard to read. One way to make them larger would be to switch row and columns, thus having three panels across for Ann, DJF and JJA, and time running downward. Similarly for Fig. 9.

p. 37, Fig. 13: It's hard to read PI values beneath the other lines. Maybe these could be shown on a vertical axis to the right of the timeline.

---

## Author Comment (AC1) · 3 Sep 2018

We would like to thank Alexander Robinson for the evaluation of our study and the constructive comments which helped to significantly improve the manuscript. Please find our responses attached as a supplement. The supplements also include a pdf showing the changes we made to the manuscript.

Please also note the supplement to this comment:
https://www.clim-past-discuss.net/cp-2018-81/cp-2018-81-AC1-supplement.zip

---

## Author Response (AR1)

**We would like to very much thank the reviewer Alexander Robinson for reviewing our study and his constructive comments which helped to significantly improve our manuscript. Please find below the reviewer's comments in black font and the author's response in blue font.**

**Responses to Alexander Robinson (Referee # 1)**

This study is focused on understanding the challenges and sources of uncertainty of simulating the surface mass balance (SMB) of the Eemian interglacial period. Steady-state time slice simulations are performed for the Eemian and the present day, with global and regionally downscaled climatic forcing applied to several combinations of SMB models. The manuscript does a good job of describing many aspects of Eemian smb modeling that are often overlooked (seasonal changes in climate, sea-ice extent, lapse rate validation). The review of past Eemian sea-level contribution estimates is also well done, even if it is only part of the motivation for the current work rather than the main focus. I think that the paper should be published after minor revisions, explained below.

We thank you for your overall positive evaluation of our study and hope that we address your comments in the following paragraphs to your satisfaction.
* * *
This is an excellent time-slice study with a good experimental design and thorough analysis. However, it is missing any insight into the role of feedbacks in transient coupled experiments where the ice-sheet topography could evolve. This could arguably be as important as the inherent bias that a particular smb model imposes, or even more so – see Robinson and Goelzer (2014), for example. I suggest adding some discussion of this point (note that this is a different point than that of the last paragraph on Page 18, and the first paragraph of Page 19 is more focused on whether a given time slice is realistic).

We agree with you and acknowledge that we failed to discuss this very important issue. The following paragraph was added to the discussion section:

"Furthermore, Ridley et al. (2005) find an additional surface warming in Greenland in transient coupled 4xCO2 ice sheet-GCM simulations compared to uncoupled simulations caused by an albedo-temperature feedback. Similarly, Robinson and Goelzer (2014) show that 30% of the additional insolation-induced Eemian melt is caused by the albedo-melt feedback. Somewhat unexpectedly, given the higher temperatures, Ridley et al. (2005) find more melting in stand-alone ice sheet simulations than in the coupled simulations. The local climate change in the coupled runs results in a negative feedback that likely causes reduced melting and enhanced precipitation. They propose the formation of a convection cell over the newly ice-free margins in summer which causes air to rise at the margins and descent over the high-elevation ice sheet (too cold for increased ablation). This leads to stronger katabatic winds which cool the lower regions and prevent warm air from penetrating towards the ice sheet. An increased strength of katabatic winds can also be caused by steeper ice sheet slopes (Gallée and Pettré, 1998; Le clec'h et al., 2017)."

added references:
Gallée, H. and Pettré, P.: Dynamical Constraints on Katabatic Wind Cessation in Adélie Land, Antarctica, Journal of the Atmospheric Sciences, 55, 1755–1770, https://doi.org/10.1175/1520-0469(1998)055<1755:DCOKWC>2.0.CO;2, 1998.

Ridley, J. K., Huybrechts, P., Gregory, J. M., and Lowe, J. A.: Elimination of the Greenland Ice Sheet in a High CO2 Climate, Journal of Climate, 18, 3409–3427, https://doi.org/10.1175/JCLI3482.1, 2005.
Robinson, A. and Goelzer, H.: The importance of insolation changes for paleo ice sheet modeling, The Cryosphere, 8, 1419–1428, http://doi.org/10.5194/tc-8-1419-2014, 2014.
* * *
While I found the analysis very thorough, it was difficult to agree with the overall conclusions reached by the authors. For example, I disagree with this sentence from the abstract: "We suggest that future Eemian climate model inter-comparison studies are combined with different SMB models to quantify Eemian SMB uncertainty estimates." To me this is a strange conclusion to make, or perhaps I don't understand the phrasing clearly. Should we believe PDD is providing added information to an energy balance model? This also comes up in the last paragraph of the Discussion. The authors seem to conclude that all SMB models are needed, because emissivity of the atmosphere is uncertain. This is a strong conclusion, but here nothing was done with emissivity. Further, wouldn't a more prudent conclusion be that deeply uncertain parameters in complex models should include sensitivity experiments (parameter perturbation) rather than simply reverting to simpler models known to lack important processes?

We agree that our conclusions were not well phrased in this regard. We wanted to make the point that it is important to also have a scheme in place to capture SMB uncertainty. We rephrased in the different parts of the paper as follows:

abstract:
"We suggest that future Eemian climate model  intercomparison studies should include SMB estimates and a scheme to capture SMB uncertainties."

discussion section:
"Since it is not feasible to perform transient fully-coupled climate-ice sheet model runs with several regional climate models, it is desirable to perform Eemian ice sheet simulations within a model intercomparison covering a range of different  climate forcings (ideally finer than 1° to capture orographic precipitation and narrow ablation zones). However, it is also essential to capture SMB uncertainties in such a model intercomparison. This could for example be realized by employing several SMB models and/or by performing sensitivity experiments of highly uncertain SMB model parameters (e.g., emissivity or melt factors). For the early Eemian it appears to be essential that the used SMB models include shortwave radiation. Furthermore, if lower resolution global climate is used, it might be worth to investigate options for correcting not just the temperature, but also the precipitation/accumulation fields."

conclusion section:
"To improve the Eemian SMB estimate,  enhanced efforts are needed in developing fully-coupled  climate-ice sheet models  efficient enough to be run over

 glacial timescales (~100 kyr), capturing the evolution of the interglacial as well as the preceding glacial ice sheets and the corresponding surface and topography changes (both are essential for estimating the Eemian sea level rise contribution). These coupled climate model runs could be downscaled at key time steps covering the Eemian period with a regional climate model, providing more accurate SMB estimates. In a next step, intermediate models like BESSI, could be used to provide SMB uncertainty estimates of this best guess SMB via model parameter sensitivity tests. To capture the uncertainty in the simulated global climate from GCMs, it would be an advantage to include dedicated experiments in a climate model intercomparison project."
* * *
Along those lines, I think it would have been quite interesting to see if using different parameter values (for example changing the emissivity of the atmosphere), it would be possible to bracket the MAR-SEB results on both sides with MAR-BESSI (SMB at 130ka showing negative and positive anomalies). That would go a long way towards showing that lower complexity smb models can be useful, but several simulations may be necessary to sample the uncertainty. [This is only a suggestion, not a requirement for publication.]

We thank you for this interesting suggestion and acknowledge that it would be very promising to use BESSI to estimate SMB uncertainties. Furthermore, BESSI could also be forced with transient climate simulations instead of steady-state simulations in the future. However, BESSI is in active development and once the identified shortcomings, i.e., the simple albedo scheme, are improved, BESSI will be a valuable tool to be tested in more paleo applications. We therefore keep your suggestion in mind for future studies.
* * *
Generally, the manuscript could use a revision for English usage as well. Some mistakes are highlighted below. Particularly, I noticed the article "the" missing in many instances.

We have reviewed the manuscript again for English usage, and rephrased and simplified many formulations.

== Minor comments =====
Page 1, line 2: Eemian interglacial => Eemian interglacial period
This formulation has been changed throughout the whole manuscript.

Page 1, line 5: "introduces uncertainties" sounds a bit strange, consider rephrasing.
This was changed accordingly.

Page 1, line 10: the calculation of insolation should be straightforward – do you mean shortwave radiation at the surface?
We wanted to say that it is important whether insolation is included in the SMB model or not and we acknowledge that it was not formulated well. It has been rephrased as follows:
"For the relatively warm early Eemian, the differences between SMB models are large which is associated with  whether insolation is included in the respective models."

Page 1, line 12: simulated climate => simulated climate,
This was changed accordingly.

Page 2, line 1: Past interglacials => Past interglacial periods [Generally this should be changed throughout, as "interglacial" is only and adjective.]
Page 2, line 7: pre-industrial => pre-industrial period
This was changed throughout the manuscript.

Page 4, line 30: surface air temperature => near-surface air temperature [?]
Yes, we mean near-surface air temperature. It was changed accordingly.

Page 4, line 34: "The only process it neglects" <= This is a strong statement, consider rephrasing.
We agree this was formulated to strongly. It was rephrased as follows:
" However, it neglects sublimation which is of low importance for the mass balance of Greenland."

Page 5, table 1: Units of PDD factors should be "mm/K/day"
Page 6, line 15: linearly => bilinearly [?]
Page 6, line 7: This 30 years => These 30 years
This was changed accordingly.

Page 8, Fig. 1: Lighter colors in the lower boxes would make this figure easier to read.
The figure was revised with lighter colors.

Page 10, line 13: "with an adapted PDD scheme" <= the ITM equation used by Robinson et al. (2011) and Calov et al. (2015) is not a PDD scheme, it is a "linearized energy-balance" scheme (originally published by Pollard, 1980).
We apologize for this mistake and rephrased as follows:
"The exceptions are Robinson et al. (2011) and Calov et al. (2015) who use an intermediate complexity statistical downscaling with  a linearized energy-balance scheme to also include shortwave radiation."

Page 13, line 35: are we using => we use
This was changed accordingly.

Page 14, line 20: refreeze => refreezing [Change everywhere it appears as a noun.]
"Refreeze" was changed to "refreezing" throughout the manuscript.

Page 15, line 6: "warmer/cooler at 125/130 ka" <= Consider reversing the time order here for consistency with elsewhere.
We agree, reversing the time order here makes more sense and we changed it accordingly.

Page 15, line 7: I think Arctic warming and amplification are not synonymous, consider revising here somewhat for clarity.
We agree that these two phrases are not synonymous and we skipped the phrase amplification.

Page 15, line 13: During early Eemian => During the early Eemian

Page 15, line 18: Sea ice are => Sea ice is
This was changed accordingly.

Page 18, first paragraph: This seems more like Discussion than Results.
You are right, this paragraph was moved to the discussion section.

== References =====
Robinson, A. and Goelzer, H.: The importance of insolation changes for paleo ice sheet modeling, The Cryosphere, 8, 1419-1428, https://doi.org/10.5194/tc-8-1419-2014, 2014.

We thank A. Robinson again for the overall positive evaluation of our manuscript and his comments which improved our manuscript significantly!

**We would like to very much thank the anonymous referee #2 for reviewing our study and her/his constructive comments which helped to significantly improve our manuscript. Please find below the referee's comments in black font and the author's response in blue font.**

**Responses to Anonymous Referee #2**

General Comments

This paper considers sources of uncertainty in simulating Greenland ice sheet surface mass balance (SMB) during the Eemian interglacial. The authors use a global Earth System model (NorESM), a regional climate model (MAR), and three kinds of SMB model (a positive-degree-day scheme, a model of intermediate complexity, and a full surface-energy-balance model) to assess the sensitivity of Eemian SMB to climate model resolution and SMB model complexity. The authors find that for earlier Eemian time slices (130 and 125 ka, with high summertime Northern Hemisphere insolation), results are sensitive to model choices, with regionally-forced SMB models giving a more negative SMB than globally-forced models, and with the PDD model underestimating melting compared to the more complex models. For later Eemian time slices (120 and 115 ka, with lower insolation), the SMB model is less critical, but SMB remains sensitive to the resolution of the forcing climate model.

The study is well designed, using a novel combination of models to draw useful inferences about SMB sensitivity for the Eemian. The authors give a broad review of earlier work and clearly describe their experimental methods. The analysis is clear and detailed, and the conclusions (with exceptions noted below) are generally well supported by the text and figures.

My main concern is that some of the conclusions are not well supported by the simulation results. I would suggest rewriting or removing some of these statements, as described below. Also, the text would benefit from some editing for English grammar; see Technical Corrections. Otherwise, the authors provide a solid and useful analysis of Eemian SMB sensitivity, and I suggest publication with minor revisions.

We thank you for your overall positive evaluation of our study. We will address your comments in the following paragraphs.
* * *
Specific Comments

I suggest a modified title. The current title emphasizes the sensitivity of the Eemian SMB to SMB model choice, whereas the text suggests an equally important role for the kind of climate forcing (high-resolution RCM v. lower-resolution GCM).

We changed the title to:

"Eemian Greenland Surface Mass Balance strongly sensitive to  model choice"
* * *
p. 1, l. 14: "We suggest that future Eemian climate model inter-comparison studies are combined with different SMB models to quantify Eemian SMB uncertainty estimates." Unless I

misunderstand how "SMB model" is defined, this statement is not well supported. The text identifies three kinds of SMB model: PDD, intermediate complexity (BESSI), and full surface-energy-balance (as in MAR). The results suggest that PDD schemes are inappropriate for the early Eemian, when insolation differed markedly from present-day. While BESSI results are closer to MAR, I don't see an argument that BESSI results are in any way more accurate or credible than MAR results. I would infer that future studies should use MAR-SEB or a comparable scheme, in order to minimize uncertainties. More generally, one should always use the most realistic, best validated model that is computationally practical, unless it can be shown that running a simpler, cheaper model yields closely similar results.

There are many other sources of uncertainty for simulated Eemian SMB, notably the absence in this study (as the authors point out) of time-varying topography. It seems more fruitful for future studies to explore other sources of uncertainty rather than revisit simple SMB models.

We agree, this was not well formulated. We wanted to make the point that it is important to account for SMB uncertainty. SEB models are very expensive and it is likely unfeasible to do uncertainty estimates with such kinds of models for millennial time scales. An intermediate model like BESSI could be used in combination with a SEB model to provide uncertainty estimates in future studies. The mentioned sentence was rephrased as follows:

"We suggest that future Eemian climate model  intercomparison studies should include SMB estimates and a scheme to capture SMB uncertainties."

Similar sentences in the discussion and conclusion section were rephrased in a similar fashion.

———————————————————————————————————————————————

p. 2 l. 35: "the amplification of summer warming over Greenland has been found to be effective". I'm not sure what is meant; effective for what?

This was not well formulated. The point is that increased insolation and increased GHGs cause comparable warming over Greenland. The sentence was rephrased to make the point of the cited study clearer:

"Furthermore,  Masson-Delmotte et al. (2011) find a similar Arctic summer warming over Greenland with the higher Eemian insolation as for a future doubling of atmospheric $CO_2$ given fixed pre-industrial insolation."

———————————————————————————————————————————————

p. 3, l. 2: Overall, I found Section 2 to be a very clear and helpful description of the models and methods.

Thank you.

———————————————————————————————————————————————

p. 4, l. 33: "The only process it neglects. . ." I suggest "It neglects. . .", since there are bound to be other neglected processes.

The sentences was rephrased as follows:

" However, it neglects sublimation which is of low importance for the mass balance of Greenland."

———————————————————————————————————————————————

p. 6, ll. 24ff: When I read this the first time, I wondered whether the study used the same static surface topography for each time slice. It does, as stated later, but I suggest stating it here.

We agree, it is important to mention this before the discussion. The following sentence was added:

"All climate simulations in this study use a static pre-industrial ice sheet."
* * *
p. 9, ll. 1ff: I liked the comprehensive description of earlier studies and their limitations. However, this section might fit better into the overall structure if swapped with Section 2.

We swapped the background section with section 2 as suggested. Please note that Fig. 4 (now Fig. 2) was revised.
* * *
p. 9, l. 4: A broad range of 0.4 to 5.6 m is given, but the more recent studies have an upper bound of ~3 m. Does this narrowing of the range (combined with the more recent ice core evidence) suggest that the high-end estimates likely are too high?

Robinson et al. 2011 provide one of the highest estimates and they use rather recent paleo constraints. They perform a large ensemble of simulations and sort out simulations which do not fit constraints of surface change and peak temperature at GRIP. And they conclude that their highest estimates are the most likely because these are the simulations which come closest to the reconstructed peak temperature at GRIP. We are not sure how results from the NEEM ice core would influence these results.

Another issue is how sea level rise (SLR) is calculated. The recent studies with the highest SLR, Robinson et. al 2011 and Born and Nisancioglu 2012, are also the ones with the largest simulated pre-industrial ice sheets. The drop between the simulated pre-industrial and minimum Eemian ice sheet is large and it makes a big difference if you calculate the SLR as a ratio between pre-industrial and Eemian (assuming some SLR value, e.g., 7m, for pre-industrial) or if you take the actual ice volume decrease and spread it evenly over the ocean. In case of Robinson et. al 2011 the difference between these two calculations is more than 1m!

In conclusion, we don't think you can say that the most likely upper bound is 3m.
* * *
p. 10, l. 33: Please say what is meant by "model consistent", or otherwise give a bit more detail about how the 3D lapse rate is computed.

We added an explanation. The sentence now reads:

"The NorESM temperature is bilinearly interpolated to the MAR grid and corrected to the MAR topography with a model consistent, temporally and spatially varying lapse rate derived from NorESM, i.e, we use the lapse rate of the NorESM atmosphere above each grid cell."
* * *
p. 13, l. 15: "the ablation in the SW reaches much lower values". Please clarify whether ablation is lower, or the SMB is lower (i.e., more negative).

This was phrased wrongly and is now clarified as follows:

"Furthermore, the  SMB in the SW  is much more negative than our reference MAR-SEB results."
* * *
p. 15, l. 2: Can you say why the annual warming signal is less pronounced in NorESM? Do you suspect a winter cold bias, a summer cold bias, or both? (I don't think this is critical to explain, just helpful if you can make an educated guess. Similarly for the next question.)

This is a good point. Unfortunately it is not entirely clear to us why the simulated warming in the northern high latitudes in early Eemian are weaker compared to the other model simulations (e.g. Lunt et al., 2013). The historical simulation of NorESM shows a positive DJF and annual temperature bias in the Arctic and a negative JJA temperature bias. However, this cannot be directly linked to the explanation of the simulated weak Arctic warming during early Eemian.

We suspect that the simulated sea ice (both extent and thickness) might be greater compared to other simulations, which can play an important role in the simulated temperature response.
It is also very likely that the model under-/over-estimate certain feedback processes in the Arctic region. As we don't have any definite answer to the comments, we'd prefer to keep the main text concise without mentioning these speculations.

Lunt, D. J., Abe-Ouchi, A., Bakker, P., Berger, A., Braconnot, P., Charbit, S., Fischer, N., Herold, N., Jungclaus, J. H., Khon, V. C., Krebs-Kanzow, U., Langebroek, P. M., Lohmann, G., Nisancioglu, K. H., Otto-Bliesner, B. L., Park, W., Pfeiffer, M., Phipps, S. J., Prange, M., Rachmayani, R., Renssen, H., Rosenbloom, N., Schneider, B., Stone, E. J., Takahashi, K., Wei, W., Yin, Q., and Zhang, Z. S.: A multi-model assessment of last interglacial temperatures, Clim. Past, 9, 699–717, https://doi.org/10.5194/cp-9-699-2013, 2013.
* * *
p. 15, l. 7: Are you able to explain why Arctic amplification is mostly absent in the early Eemian?

We are afraid that we are unable to explain the absence of the amplification in the early Eemian. Please note that we rephrased the sentence to be more concise:

"Arctic warming  is absent, or not pronounced in both seasons in the early Eemian."
* * *
p. 18, l. 2: I think "challenging" is not the right word here; not including SW for the Eemian seems like a more fundamental flaw. Maybe "highly problematic"? Similarly, "challenges" in l. 9 below could be replaced with something like "complicates".

We agree that problematic is a better word here. Please, note that the whole paragraph was moved to the discussion section. The sentence now reads:

"Firstly, it is problematic not to include shortwave radiation in a SMB model when investigating the Eemian, because the melt might be underestimated."

The second sentences was changed to:

"This complicates PDD-derived Eemian SMB estimates since insolation is included in PDD models."

A similar phrase in the abstract was also changed.
* * *
p. 18, l. 4: I suggest removing "or other deficiencies", since deficiencies apart from coarse resolution haven't been discussed.
We removed this part.
* * *
p. 18, l. 11: I'm not sure NorESM should be described as "relatively high resolution". Its resolution is low compared to MAR, and is not high compared to other IPCC-class ESMs. Some global ESMs, for example, run with a 1 degree rather than 2 degree atmosphere.
We removed this description here and further down in the conclusion.
* * *
Section 6: The first part of the discussion appropriately focuses on big issues such as variable topography and climate forcing resolution. Later, e.g. the second full paragraph on p. 20, it gets into finer details such as refreezing and temporal resolution in BESSI, which might fit better in Section 5.
We acknowledge your concern. However, we prefer to keep the paragraphs in the discussion section.
* * *
p. 18, l. 27: A discussion of evolving Greenland topography should refer to the study of Ridley et al. 2005 (in the context of future warming and deglaciation), and possibly some more recent coupled ESM-ISM studies.
We added a paragraph to the discussion section:
"Furthermore, Ridley et al. (2005) find an additional surface warming in Greenland in transient coupled 4xCO2 ice sheet-GCM simulations compared to uncoupled simulations caused by an albedo-temperature feedback. Similarly, Robinson and Goelzer (2014) show that 30% of the additional insolation-induced Eemian melt is caused by the albedo-melt feedback. Somewhat unexpectedly, given the higher temperatures, Ridley et al. (2005) find more melting in stand-alone ice sheet simulations than in the coupled simulations. The local climate change in the coupled runs results in a negative feedback that likely causes reduced melting and enhanced precipitation. They propose the formation of a convection cell over the newly ice-free margins in summer which causes air to rise at the margins and descent over the high-elevation ice sheet (too cold for increased ablation). This leads to stronger katabatic winds which cool the lower regions and prevent warm air from penetrating towards the ice sheet. An increased strength of katabatic winds can also be caused by steeper ice sheet slopes (Gallée and Pettré, 1998; Le clec'h et al., 2017)."

added references:
Gallée, H. and Pettré, P.: Dynamical Constraints on Katabatic Wind Cessation in Adélie Land, Antarctica, Journal of the Atmospheric Sciences, 55, 1755–1770, https://doi.org/10.1175/1520-0469(1998)055<1755:DCOKWC>2.0.CO;2, 1998.
Ridley, J. K., Huybrechts, P., Gregory, J. M., and Lowe, J. A.: Elimination of the Greenland Ice Sheet in a High CO2 Climate, Journal of Climate, 18, 3409–3427, https://doi.org/10.1175/JCLI3482.1, 2005.
Robinson, A. and Goelzer, H.: The importance of insolation changes for paleo ice sheet modeling, The Cryosphere, 8, 1419–1428, http://doi.org/10.5194/tc-8-1419-2014, 2014.
* * *
p. 19, l. 5: "neglecting the meltwater influx to the ocean from the retreating glacial ice gives warmer simulated air temperatures". Can you say briefly why this is the case?

We added a short explanation in brackets:

"Additionally, neglecting the meltwater influx to the ocean from the retreating glacial ice sheet gives warmer simulated temperatures (the light meltwater would form a fresh surface layer on the ocean and isolate the warm sub-surface water from the atmosphere)."
* * *
p. 21, l. 1: "it is hard to argue why a energy balance model which needs poorly constrained information (e.g., net radiation) would produce more reliable results for paleo ablation than a simple PDD model". I don't think this statement is well supported For example, incoming solar insolation is very well constrained by orbital calculations, and this alone is a good reason that an energy-balance model might produce more reliable paleo ablation than a PDD model.

We agree that incoming solar insolation is well constrained. However, outgoing, and notably the ratio of incoming longwave radiation because of clouds is uncertain. We reformulated the sentences:

"However, in the absence of well-constrained input data, the additional complexity of more comprehensive models may be disadvantageous to the uncertainty of the simulation."
* * *
p. 21, l. 15: "different SMB models should be included in Eemian ice sheet simulations to capture uncertainties". I disagree with this statement. It is true that there will always be some uncertainties in atmospheric variables (such as cloud cover) that influence the surface energy balance. But it does not follow that "the uncertainty of Eemian global climate simulations cannot be narrowed down further." (For instance, one could build a better cloud model.). Also, I see no reason not to use the best computationally affordable SMB model (either MAR's SEB model or something comparable). See comments above for p. 1, l. 14.

p. 21, l. 18: "it is desirable to perform Eemian ice sheet simulations within a model intercomparison covering a range of different (high resolution) climate forcings and a range of SMB models". Please define what is meant by high resolution. E.g., finer than 1 degree? Fine enough to capture orographic precipitation and narrow ablation zones? I'm again unclear on the value of a range of SMB models for UQ, unless the range includes other models with SEB schemes comparable to MAR (e.g., RACMO). Also, it could be valuable to explore a range of parameter settings within MAR, to the extent that certain parameters are uncertain and tunable.

This section was revised as follows:

"However, it remains challenging to quantify the uncertainty contributions related to global climate forcing (not tested here) and to SMB model choice. More sophisticated SMB models might seem like  an obvious choice for future studies of the Eemian Greenland ice sheet due to their advanced representation of atmospheric and surface processes. However,  the uncertainty of Eemian global climate simulations  will always play an important role for SMB calculations in paleo applications (e.g., cloud cover and other poorly constrained atmospheric variables

capture uncertainties related to model selection in paleo applications). Since it is not feasible to perform transient fully-coupled climate-ice sheet model runs with several regional climate models, it is desirable to perform Eemian ice sheet simulations within a model intercomparison covering a range of different (high resolution) climate forcings and a range of SMB models to capture uncertainties in the best possible way climate forcings (ideally finer than 1° to capture orographic precipitation and narrow ablation zones). However, it is also essential to capture SMB uncertainties in such a model intercomparison. This could for example be realized by employing several SMB models and/or by performing sensitivity experiments of highly uncertain SMB model parameters (e.g., emissivity or melt factors). For the early Eemian it appears to be essential that the SMB models include shortwave radiation. Furthermore, if lower resolution global climate is used, it might be worth to investigate options for correcting not just the temperature, but also the precipitation/accumulation fields."
* * *
p. 22, l. 1: "we recognize that a further improved intermediate complexity SMB model (i.e. albedo parameterization) would be very useful for forcing ice sheet models on paleo time scales." I agree that models like BESSI could be improved for paleo simulations, but I don't see why an improved intermediate model would be preferable to SMB forcing from a detailed RCM. Assuming that you're already using MAR or another RCM for dynamical downscaling, why not just use the RCM's SMB?
BESSI and other intermediate models could for example be used to provide uncertainty estimates of the SEB-derived SMB (especially for long timescales and transient simulations), by performing parameters sensitivity tests, because it is challenging to run ensembles with an SEB model over long timescales.
* * *
p. 22, l. 7: "further effort needs to be put in developing fully-coupled regional climate-ice sheet models and making them efficient enough to be run over whole glacial interglacial cycles". I'm unclear on the role of global models here. Is the idea that the RCM would be run interactively with a global climate model, or just the ice sheet model? Also, what is meant by a whole glacial-interglacial cycle? Do you mean an interglacial time scale (~10 kyr) or a full glacial time scale (~100 kyr)?

I think that coupled GCM-ISMs have a role to play, which is not acknowledged here. Other GCMs/ESMs could prove to be more accurate than NorESM for Eemian SMB studies, using some combination of higher (or spatially variable) resolution, improved cloud and snow physics, and SEB schemes with subgrid elevation classes. Even if the SMB from a global ESM is less accurate than the dynamically downscaled SMB from MAR, this disadvantage could be offset by the benefits of simulating topographic feedbacks in a global model.

p. 22, l. 8: I disagree with the last sentence of the conclusions (in particular, "combining with various SMB models"), for the reasons stated above.
We acknowledge that this idea was not thought all the way through. A coupled system with global and regional climate model and an ice sheet model is probably unfeasible for still some years. We revised the section and formulated our suggestion differently:
"To improve the Eemian SMB estimate, further effort needs to be put enhanced efforts are needed in developing fully-coupled regional climate-ice sheet models and making them

efficient enough to be run over  glacial timescales (~100 kyr), capturing the evolution of the interglacial as well as the preceding glacial ice sheets and the corresponding surface and topography changes (both are essential for estimating the Eemian sea level rise contribution). These coupled climate model runs could be downscaled at key time steps covering the Eemian period with a regional climate model, providing more accurate SMB estimates. In a next step, intermediate models like BESSI, could be used to provide SMB uncertainty estimates of this best guess SMB via model parameter sensitivity tests. To capture the uncertainty in the simulated global climate from GCMs, it would be an advantage to include dedicated experiments in a climate model intercomparison project."
* * *
Technical Corrections

p.2 l. 10: "While" -> "However"
p. 2 l. 17: "Global Circulation Models" -> "Global Climate Models"
p. 2, l. 19: No caps in Surface Mass Balance. Likewise Surface Energy Balance, l. 24
p. 2, l. 27: Delete "due to"
p.2, l. 28: "which is the reason for" -> "which are primarily responsible for"
The text was changed according to these comments.

p. 4, l. 27: typo, "Ber/ge/n"
Has been changed to BErgen Snow SImulator (BESSI). The model was developed in Bern and is now developed further in Bergen.

p. 5, l. 2: "Firn densification is realized with models. . .". Awkward phrasing; please reword.
We rephrased to:
"Firn densification is simulated with models commonly used in ice core research,..."

p. 7, l. 6: "This 30 years" -> "These 30 years of output. . ."
p. 7, l. 8: "downwards "-> "downward"
p. 8, l. 14: Add comma after "topography"
p. 9, l. 14: Delete "a" before "Eemian"
p. 10, l. 4: "we are not discussing the ice dynamics used further." Suggest "we do not further discuss the ice dynamics."
p. 11, Fig. 3: "Nisancioglu" is misspelled. Suggest adding "Simulated" before "sea level rise" in the title.
p. 12, l. 9: Add units after "5"
p. 13, l. 6: No commas needed in this sentence.
p. 13, l. 22: No comma after "Both"
p 13, l. 29: "lower-resolution" (with a hyphen)
p. 13, l. 35: "are we using" -> "we are using"
p. 14, l. 9: can not -> cannot
p. 15, l. 14: "with ice thickness thinner" -> "with ice thinner"
The text was changed or rephrased according to these comments.

p. 15, l. 18: "is thicker" -> "are thicker". Also, do you mean an ice thickness increase?

Yes, you are right, it is an ice thickness increase.

p. 17, l. 11: Delete "the" before "their"
p. 18, l. 4: Suggest "Both the climate and the type of SMB are important"
p. 18, l. 30: Misplaced parentheses for Merz citation
p. 19, l. 6: "assumed" is not the right word, since you've given an argument. Suggest ". . .130 ka temperatures are likely warmer than the actual temperatures, resulting in. . ."
p. 19. l. 24: No quotes needed for "cooler climate states". Likewise below for "warmer climate states".
p. 21, l. 7: "assumption" -> "inference"
p. 21, l. 32: No comma needed after "Despite"

The text was changed or rephrased according to these comments.

p. 29, Fig. 5: Use the same symbol for, e.g., ice cores in both temperature and precipitation plots. Should l. 2 of the caption read "temporally and spatially varying 3d lapse rate"?

The symbols have been switched and the caption has been corrected.

p. 32, Fig. 8: The panels are small and hard to read. One way to make them larger would be to switch row and columns, thus having three panels across for Ann, DJF and JJA, and time running downward. Similarly for Fig. 9.

We acknowledge your concern about the small figure panels. However, we prefer to leave the figures as they are, because we like to facilitate comparison with our previous simulations performed by the lower resolution of NorESM (e.g. Fig. 2 in Langebroek and Nisancioglu, 2013) where a similar figure format was used.

Langebroek, P. M. and Nisancioglu, K. H.: Simulating last interglacial climate with NorESM: role of insolation and greenhouse gases in the timing of peak warmth, Clim. Past, 10, 1305-1318, https://doi.org/10.5194/cp-10-1305-2014, 2014.

p. 37, Fig. 13: It's hard to read PI values beneath the other lines. Maybe these could be shown on a vertical axis to the right of the timeline.

We added it as additional triangles on the right side.

We thank the anonymous referee again for the overall positive evaluation of our manuscript and his comments which improved our manuscript significantly!

[revised manuscript text omitted]